# Inactivation of the ATMIN/ATM pathway protects against glioblastoma formation

Sophia M Blake[1,2†], Stefan H Stricker[3‡], Hanna Halavach[1,2], Anna R Poetsch[2,4,5,6], George Cresswell[2,4], Gavin Kelly[2,7], Nnennaya Kanu[1,2§], Silvia Marino[8], Nicholas M Luscombe[2,4,5,6], Steven M Pollard[3¶], Axel Behrens[1,2,9*]

[1]Adult Stem Cell Laboratory, The Francis Crick Institute, London, United Kingdom; [2]Lincoln's Inn Fields Laboratory, The Francis Crick Institute, London, United Kingdom; [3]Samantha Dickson Brain Cancer Unit and Department of Cancer Biology, UCL Cancer Institute, University College London, London, United Kingdom; [4]Bioinformatics and Computational Biology Laboratory, The Francis Crick Institute, London, United Kingdom; [5]UCL Genetics Institute, Department of Genetics, Evolution and Environment, University College London, London, United Kingdom; [6]Okinawa Institute of Science and Technology, Okinawa, Japan; [7]Bioinformatics and Biostatistics, The Francis Crick Institute, London, United Kingdom; [8]Blizard Institute, Barts and the London School of Medicine and Dentistry, London, United Kingdom; [9]Faculty of Life Sciences and Medicine, King's College London, London, United Kingdom

*For correspondence: axel.
behrens@crick.ac.uk

Present address: †Department of Pharmacology and Translational Research, Boehringer Ingelheim RCV GmbH & Co KG, Vienna, Austria; ‡Helmholtz Zentrum München, German Research Center for Environmental Health, Institute of Stem Cell Research, Neuherberg, Germany; §Translational Cancer Therapeutics Laboratory, UCL Cancer Institute, University College London, London, United Kingdom; ¶MRC Centre for Regenerative Medicine, The University of Edinburgh, Edinburgh, United Kingdom

Competing interests: The authors declare that no competing interests exist.

**Abstract** Glioblastoma multiforme (GBM) is the most aggressive human primary brain cancer. Using a *Trp53*-deficient mouse model of GBM, we show that genetic inactivation of the Atm cofactor Atmin, which is dispensable for embryonic and adult neural development, strongly suppresses GBM formation. Mechanistically, expression of several GBM-associated genes, including *Pdgfra*, was normalized by *Atmin* deletion in the *Trp53*-null background. Pharmacological ATM inhibition also reduced *Pdgfra* expression, and reduced the proliferation of *Trp53*-deficient primary glioma cells from murine and human tumors, while normal neural stem cells were unaffected. Analysis of GBM datasets showed that *PDGFRA* expression is also significantly increased in human *TP53*-mutant compared with *TP53*-wild-type tumors. Moreover, combined treatment with ATM and PDGFRA inhibitors efficiently killed *TP53*-mutant primary human GBM cells, but not untransformed neural stem cells. These results reveal a new requirement for ATMIN-dependent ATM signaling in TP53-deficient GBM, indicating a pro-tumorigenic role for ATM in the context of these tumors.

## Introduction

Glioblastoma multiforme (GBM) is the most common and aggressive form of primary brain cancer in adults. Despite improvements in clinical care, a rapid disease progression and insufficient understanding of the etiology of these tumors results in very poor survival prognosis. The standard treatment for GBM patients involves a combination of the DNA damaging agent temozolomide together with radiotherapy (*Stupp et al., 2005*). However, the benefits of current treatment regimes come with severe side effects for patients, as well as drug resistance and inevitable recurrence of the tumor (*Wen and Kesari, 2008*).

Over recent years, great efforts by individual research groups and consortia have shed light on the key genetic events that lie at the heart of human GBM formation, raising the possibility of a more targeted approach to therapy. This has identified activation of growth factor receptor signaling

**eLife digest** Glioblastomas are the most common and aggressive brain cancers in adults, and currently lack efficient treatment options. Glioblastoma cells contain genetic mutations that enable them to grow and divide more quickly than they would under normal conditions. The occurrence of these mutations often leads to a functional impairment in so-called 'tumor suppressor' proteins that may have a range of roles, including repairing genetic damage or controlling the rate of cell division.

Blake et al. have now studied how some of these tumor suppressor proteins interact. Deleting a prominent tumor suppressor called TP53 from the brain of mice caused these animals to develop glioblastomas. If, however, both TP53 and another tumor suppressor called ATMIN were deleted at the same time, the majority of mice did not develop any brain tumors. Further in-depth profiling of these brain tumor cells revealed that TP53-deleted cells had very high levels of the oncogene PDGFRA, which causes cells to divide more rapidly. These high PDGFRA levels were brought back to normal conditions upon deletion of ATMIN.

Blake et al. then studied primary human glioblastoma cells that lack TP53 and found that these cells could be efficiently killed by a combination of drugs that block the activity of PDGFRA and the protein ATM, which is known to work in concert with ATMIN. Importantly, this combination of drugs did not adversely affect healthy brain cells, opening up new strategies and potential treatment options for glioblastoma patients.

(for example by amplification or overexpression of the PDGF receptor) and direct or indirect inactivation of the TP53 and retinoblastoma tumor suppressors as core deregulated pathways in human GBM (*Cancer Genome Atlas Research Network, 2008*). Unfortunately, many of these pathways are difficult to target pharmacologically in vivo, and for others, like PDGFR signaling, available inhibitors have so far lacked clinical impact (*Wen et al., 2006*; *Rich et al., 2004*). Hence, an important goal of GBM research is to identify novel, more effective therapies to generate better outcomes.

One of the central determinants of tumor progression and the response to therapy is the DNA damage response (*Lord and Ashworth, 2012*). The DNA damage kinase ATM is known primarily as a tumor suppressor, through its role in the response to DNA double-strand breaks (*Shiloh and Ziv, 2013*), and systemic loss of ATM has previously been shown to accelerate glioblastoma progression (*Squatrito et al., 2010*). This tumor suppressive role is in line with the activation of the DNA damage response in precancerous lesions as a barrier to tumorigenesis (*Bartkova et al., 2005*). A recent study has, however, demonstrated loss of ATM signaling to inhibit the growth of TP53-null tumor xenografts, via stabilization of p14$^{ARF}$ (*Velimezi et al., 2013*), suggesting that ATM function in cancer is highly context-dependent.

As well as its canonical activation at break sites, ATM signaling also occurs in response to other cellular stresses (*Bakkenist and Kastan, 2003*), and this mode of ATM signaling requires the ATM INteractor ATMIN (*Kanu and Behrens, 2007*). ATMIN interacts with the ATM kinase in basal conditions and disassociates from ATM in response to ionizing radiation, to allow ATM to interact with the MRN complex at double-strand break sites (*Zhang et al., 2014*). ATMIN also has ATM-independent functions, most notably the transcriptional activation of Dynll1, a motor protein involved with ciliogenesis and crucial for lung development (*Jurado et al., 2012a*; *Goggolidou et al., 2014*). In addition, ATMIN has been shown to counteract oxidative damage in the brain (*Kanu et al., 2010*) and to protect against B cell lymphomagenesis (*Loizou et al., 2011*), but its role in other cancer types has not yet been determined.

A key downstream mediator of the DNA damage response pathway, activated by both the double-strand break and ATMIN-dependent responses, is the tumor suppressor TP53. Precancerous lesions in which the DNA damage response is activated are under selective pressure to lose or mutate *TP53*, and loss of *TP53* is known to cooperate with several genes to accelerate tumorigenesis (as summarized in the IARC database [*Petitjean et al., 2007*]). Among these cooperating changes is loss of *ATM*, which induces rapid T-cell lymphoma development (*Westphal et al., 1997*). *TP53* is also one of the most commonly mutated genes in human GBM. Consequently, the majority of available mouse models of GBM use deletion of one or both copies of the *Trp53* gene in combination

with other mutations (*Chen et al., 2012*). Overexpression of the Pdgf receptor ligand *Pdgfrb* in adult Nestin-positive neural stem cells, for instance, results in glioma formation, which is accelerated in a *Trp53*-mutant background (*Squatrito et al., 2010*).

In this study, we demonstrate that congenital loss of *Trp53* in the mouse brain is sufficient to precipitate spontaneous glioblastoma formation, and that this correlates with upregulation of *Pdgfra*. Further, we show that ATMIN plays a critical role in GBM formation, promoting Pdgfra protein and gene expression in a *Trp53*-deficient background, using an in vivo glioma model as well as neural stem cell and primary tumor cell cultures. Importantly, we find that these results are translatable to therapeutic ATM inhibition in human patient-derived GBM stem cells, and that combining ATM inhibition with PDGFRA inhibition results in synergistic tumor cell killing with minimal effects on untransformed cells.

## Results

### Glioblastomas develop at high frequency in $p53^{\Delta N}$ (*Trp53* mutant) mice

Loss of *TP53* is one of the earliest occurring events in human GBM initiation (*Maher et al., 2001*; *Wang et al., 2009*; *Ohgaki et al., 2004*; *Mazor et al., 2015*; *Johnson et al., 2014*). This inevitably results in the accumulation of a plethora of secondary hits, which, after a long latency period, leads to tumor formation. To recapitulate this chain of events in mice, we deleted *Trp53* as an initial driver during neural development (using $p53^{f/f}$; *Nestin-Cre* ($p53^{\Delta N}$) mice) and monitored brain tumor formation in late adult life. After 8 months, brain tumors arose in $p53^{\Delta N}$ mice with high penetrance (*Figure 1A and B*), similar to previous observations using an *hGFAP-Cre* model (*Wang et al., 2009*). Animals began to show neurological symptoms including reduced movements and tremor at an average age of 263 days. When examined histologically, many $p53^{\Delta N}$ tumors had features consistent with WHO classification criteria for Grade IV GBM (*Figure 1C*). Out of 17 animals, 14 (82%) showed diffusely infiltrative astrocytic brain tumors, the majority of which were classified as GBM by independent analysis (57% Grade IV, glioblastoma; 29% Grade III, anaplastic astrocytoma; 14% Grade II, fibrillary astrocytoma), while one displayed an osteosarcoma (*Figure 1B and D*, *Figure 1—figure supplement 1* and *Supplementary file 1*). $p53^{\Delta N}$ GBMs displayed pseudopalisading necrosis (i), microvascular proliferation with endothelial hyperplasia (ii) and occasional endovascular thrombosis (iii), hallmarks of human GBM tumors (*Figure 1E* [i–iii]). The markers Gfap (iv), Nestin (v), and Olig2 (vi) were expressed in all tumors, while they were negative for the neuronal marker NeuN, supporting the diagnosis of glioma (*Figure 1E* (iv–vi) and *Figure 1—figure supplement 2*). Intertumoral and intratumoral heterogeneity, a classic hallmark of high-grade gliomas, was also frequently observed (*Figure 1F*). These data indicate that early embryonic loss of *Trp*53 in the brain is sufficient to promote GBM formation. The long latency of these tumors, together with the genome instability and transcriptional changes known to be induced by loss of *Trp*53, indicated that these gliomas almost certainly develop as a result of secondary mutations arising in the *Trp*53-null brain. In line with this notion, mouse SNP array data revealed large chromosomal gains and losses (typical tumorigenic genetic changes) in four out of five analyzed $p53^{\Delta N}$gliomas, whereas no changes were detected in one wt and three Trp53-deficient non-tumorigenic control NS cells (*Figure 1—figure supplement 3*). These secondary hits are not congenitally predefined and thus possibly allow a better representation of the genetic diversity of the disease in patients. In addition, the high frequency of brain tumor formation, proportion of high-grade tumors, and consistency of tumor latency on a mixed genetic background make this a valuable GBM model.

### Loss of Atmin strongly suppresses GBM formation in the $p53^{\Delta N}$ model

As an integral part of the response to genomic stress, activation of the DNA damage response pathway typically represents a barrier to tumorigenesis. Consistent with this, it has previously been shown that loss of the DNA damage kinase Atm accelerates tumorigenesis in a Pdgf-driven glioblastoma model (*Squatrito et al., 2010*). In addition, deletion of the Atm cofactor Atmin promotes B cell lymphomagenesis (*Loizou et al., 2011*). Whether loss of Atmin affects tumorigenesis in other cancer types is so far unknown. To investigate the potential role of Atmin in glioma formation, we crossed $Atmin^{\Delta N}$ mice (*Kanu et al., 2010*) with $p53^{\Delta N}$ mice. Strikingly, $Atmin^{\Delta N}$; $p53^{\Delta N}$ mice showed significantly longer tumor-free survival than $p53^{\Delta N}$animals (p<0.0002) (*Figure 2A*). The largest

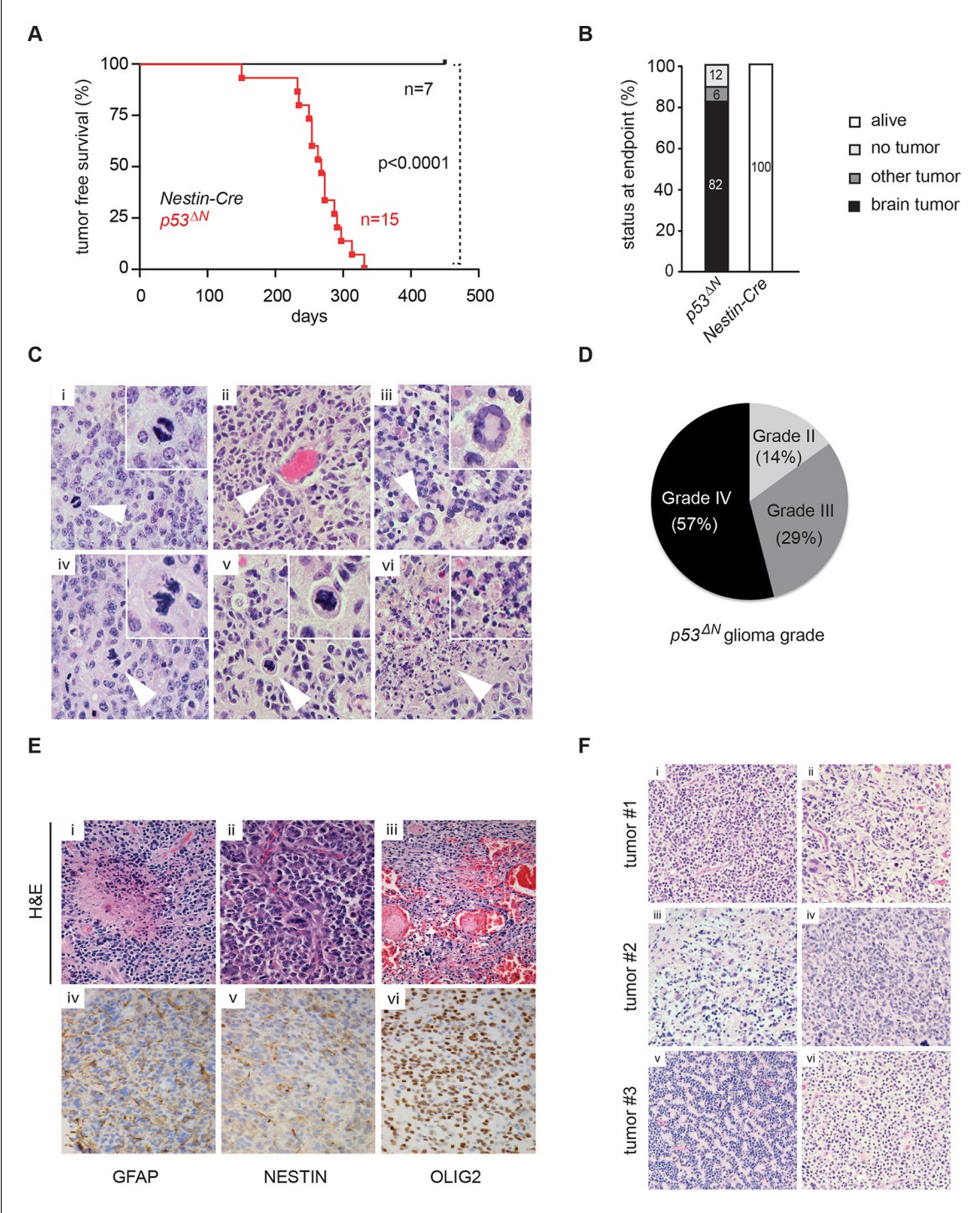

**Figure 1.** Loss of *Trp53* is sufficient to induce GBM with high penetrance. (**A**) Kaplan-Meier curves showing tumor-free survival in *p53*<sup>f/f</sup>; Nestin-Cre (*p53*<sup>ΔN</sup>) and control Nestin-Cre mice. (**B**) Status of mouse cohorts at 450 days, showing tumor incidence. (**C**) H&E-stained *p53*<sup>ΔN</sup> brain tumor sections showing histological features of malignant GBM. Arrowheads indicate (i) a mitotic cell, (ii) neo-vascularization, (iii) rosetta formation, (iv) mitotic catastrophe, (v) a multinucleated giant cell, and (vi) large areas of necrosis. (**D**) Grades of gliomas for *p53*<sup>ΔN</sup> mice. (**E**) Examples of human GBM hallmarks observed in *p53*<sup>ΔN</sup> tumors: pseudopalisading necrosis (i), microvascular proliferation with endothelial hyperplasia (ii), and endovascular thrombosis (iii) (all H&E). Immunohistochemistry shows expression of the glial markers GFAP (iv), NESTIN (v), and OLIG2 (vi) in *p53*<sup>ΔN</sup> tumors. (**F**) *p53*<sup>ΔN</sup> glioblastomas display high inter- and intra-tumoral heterogeneity. H&E images i-ii, iii-iv, and v-vi represent different regions of the same tumors #1, #2, and #3 respectively. GBM, Glioblastoma multiforme.

*Figure 1 continued on next page*

*Figure 1 continued*

The following figure supplements are available for figure 1:

**Figure supplement 1.** Histological features of lower grade tumors observed in $p53^{\Delta N}$ animals.

**Figure supplement 2.** Histological features of glioblastomas observed in $p53^{\Delta N}$ animals.

**Figure supplement 3.** SNP array analysis on primary cells derived from GBM bearing $p53^{\Delta N}$, non-tumor bearing $p53^{\Delta N}$, and wild-type (wt) animals.

**Figure supplement 4.** Table representing CGHcall output for SNP array analysis compressed to comparable regions between arrays in the series.

proportion of $Atmin^{\Delta N}$; $p53^{\Delta N}$ mice (43%) did not develop any signs of sickness and were still alive after 450 days; a further 36% succumbed either to non-tumor-related illness or, in one case, a pancreatic tumor (*Figure 2B* and *Supplementary file 1*). Only three animals out of a cohort of 14 $Atmin^{\Delta N}$; $p53^{\Delta N}$ double mutants (21%) initiated glial tumors (*Figure 2B*), compared with 82% of $p53^{\Delta N}$ mice over the same period. The three GBMs arising in $Atmin^{\Delta N}$; $p53^{\Delta N}$ double-mutant mice arose later but were histologically similar to those observed in $p53^{\Delta N}$ mice (*Figure 2—figure supplement 1*), and did not escape Atmin deletion, suggesting that the requirement for Atmin can eventually be overcome, but only in a small proportion of cases. Importantly, central nervous system deletion of Atmin alone ($Atmin^{\Delta N}$) did not affect brain morphology or histology (*Kanu et al., 2010*) and the mice remained tumor free (*Figure 2A*). Thus, Atmin deletion strongly suppresses GBM formation in $p53^{\Delta N}$ animals.

## Atmin loss rescues hyperproliferation and sensitivity to hypoxia in $p53^{\Delta N}$ NSCs

Loss of Trp53 affects diverse fundamental cellular processes including proliferation, genome stability, cell cycle arrest, and apoptosis (*Vousden and Lane, 2007*). To understand the involvement of Atmin in the suppression of the *Trp53*-null phenotype, we isolated neural stem cells (NSCs) (*Figure 2—figure supplement 2*) from $p53^{\Delta N}$ embryos and tested the effect of Atmin loss on these processes. $p53^{\Delta N}$ NSCs proliferated more rapidly than wild-type (*wt*) controls. This proliferative advantage, however, was rescued in $Atmin^{\Delta N}$; $p53^{\Delta N}$ NSCs (*Figure 2C and D*), indicating that Atmin is required for the hyperproliferation of Trp53-deficient NSCs. Loss of Atmin alone did not alter proliferation of NSCs (*Figure 2—figure supplement 3*). FACS profiling of $p53^{\Delta N}$ and $Atmin^{\Delta N}$; $p53^{\Delta N}$ NSCs revealed a decrease in the proportion of $Atmin^{\Delta N}$; $p53^{\Delta N}$ cells in S phase compared with $p53^{\Delta N}$ NSCs, in agreement with the reduced proliferation rate, but no change in cells with a >4n DNA content, and metaphase spreads at passage 10 revealed no obvious difference in ploidy (*Figure 2—figure supplement 4*). We analyzed cell cycle arrest and cell death in response to ionizing radiation (IR) by quantifying the percentage of BrdU-incorporating and DAPI-permeable NSCs respectively. As expected, $p53^{\Delta N}$ NSCs arrested less efficiently than *wt* NSCs and cell death was reduced; but cell cycle arrest and cell death were not rescued in $Atmin^{\Delta N}$; $p53^{\Delta N}$ NSCs (*Figure 2E* and *Figure 2—figure supplement 5*). In addition to IR-induced cell death, $p53^{\Delta N}$ NSCs were more resilient to hypoxia-induced death, consistent with previous studies (*Liu et al., 2007*; *Graeber et al., 1996*). Atmin deletion in these cells re-sensitized them to hypoxia, to similar levels as *wt* NSCs (*Figure 2F*). Thus, loss of Atmin is able to rescue some of the phenotypes of Trp53 loss, such as hyperproliferation and hypoxia induced death, but not others, such as IR-induced cell cycle arrest and cell death.

## Loss of Atmin impairs the tumorigenicity of neural stem cells in orthotopic transplants

As increased proliferation and hypoxia resistance are attributes commonly found in tumor initiating cells of solid tumors, including glioma (*Graeber et al., 1996*; *Gilbertson and Rich, 2007*), we evaluated the tumorigenic potential of $p53^{\Delta N}$ and $Atmin^{\Delta N}$; $p53^{\Delta N}$ NSCs in vivo. We performed intracranial injections of NSCs isolated from *wt*, $p53^{\Delta N}$, and $Atmin^{\Delta N}$; $p53^{\Delta N}$ embryos (*Figure 2G*). Five out of five animals injected with $p53^{\Delta N}$ NSCs died within 93 days post-injection, while four out of five *wt* and three out of four $p53^{\Delta N}$; $Atmin^{\Delta N}$ NSC-injected animals were still alive even at 155 days, the

endpoint of the experiment (*Figure 2G*). Histologically, injected cells appeared as atypical glial cells expressing Nestin and Ki67, diffusely infiltrating the host brain (*Figure 2—figure supplement 6*), similar to previous observations with human GBM xenografts (*Stricker et al., 2013*). Fluorescent imaging of asymptomatic mice at the experimental endpoint readily detected injected *Atmin$^{\Delta N}$*; *p53$^{\Delta N}$* and *wt* NSCs (*Figure 2—figure supplement 7*), suggesting that these cells survived in the host brain, but did not induce lethality. Thus, genetic inactivation of *Atmin* greatly impairs the tumorigenic potential of *p53$^{\Delta N}$*cells.

## Atmin loss normalizes expression of the proto- oncogene *Pdgfra* in *p53$^{\Delta N}$* NSCs

To understand the molecular basis of the observed attenuation of tumorigenic potential, we performed gene expression profiling on *wt*, *p53$^{\Delta N}$*, and *Atmin$^{\Delta N}$*; *p53$^{\Delta N}$*embryonic NSCs. Compared with *wt* NSCs, 145 genes were downregulated more than 1.5-fold in *p53$^{\Delta N}$*, whereas 77 were overexpressed (*Figure 3A*). Many of the canonical Trp53 target genes, such as *Bax*, *Puma*, and *Cdkn1a*, were downregulated to a similar extent in *Atmin$^{\Delta N}$*; *p53$^{\Delta N}$* compared with *p53$^{\Delta N}$*NSCs. However, 27% of genes deregulated in *p53$^{\Delta N}$*NSCs returned to *wt* expression levels when Atmin was also deleted (36/145 of the decreased, and 24/77 of the increased genes; examples in *Figure 3B*). When this subset of genes 'rescued' in double mutant NSCs was queried against the Cancer Genome Atlas (TCGA) database, in several cases abnormal levels of these transcripts were found to be associated with human GBM (*Cancer Genome Atlas Research Network, 2008*; *Suvasini et al., 2011*). Particularly notable among this list was the platelet-derived growth factor receptor alpha (Pdgfra) (*Figure 3C*). Elevated levels of *PDGFRA* have been observed in human gliomas of various malignancy grades (*Engström et al., 2012*), and increased PDGFR signaling has been shown to induce glioma-like growths in vivo (*Jackson et al., 2006*). Increased PDGFRA levels are also a characteristic hallmark of the proneural GBM subtype as classified in (*Brennan et al., 2013*; *Verhaak et al., 2010*). Interestingly, we found that *p53$^{\Delta N}$*NSCs were associated most closely with the proneural subtype when comparing the microarray expression profile from the murine NSCs to that of publically available human GBM samples from TCGA (*Verhaak et al., 2010*) (*Figure 3—figure supplement 1*), which is in agreement with elevated *Pdgfra* mRNA as well as protein levels (*Figure 3D*). This increased protein expression was also apparent in high-grade GBMs from endstage *p53$^{\Delta N}$*animals (*Figure 3E* and *Figure 3—figure supplement 2*). Other growth factor receptors, like Egfr, and proteins commonly deregulated in glioma including Rb, Pten, Cyclin D2, Junc, and Cdk4 (*Wang et al., 2009*) remained unaltered in *p53$^{\Delta N}$*cells and tumors (*Figure 3—figure supplements 2* and *3*), as did the closely related Pdgf receptor *Pdgfrb* and the receptor ligands *Pdgfa* and *Pdgfb* (*Figure 3—figure supplement 4*).

Importantly, deletion of *Atmin* together with *Trp53* completely rescued the elevated *Pdgfra* expression, reducing it to wild-type levels (*Figure 3C and D*). Overexpression of tagged Atmin in double mutant NSCs restored elevated Pdgfra expression at both the mRNA and protein levels (*Figure 3F and G*), while acute silencing of *Atmin* in *p53$^{\Delta N}$*NSCs, using two independent shRNA constructs, reduced *Pdgfra* levels (*Figure 3H*), implying that sustained *Pdgfra* upregulation in *p53$^{\Delta N}$*NSCs constantly requires Atmin function to maintain increased *Pdgfra* expression. Thus, interference with Atmin function is sufficient to reduce *Pdgfra* overexpression in *Trp53*-deficient cells.

To assess whether Trp53 null tumors require sustained *Pdgfra* overexpression, we isolated primary tumor cells from GBM-bearing *p53$^{\Delta N}$*mice and induced stable knockdown of *Pdgfra* using shRNA (*Figure 3—figure supplement 5*). This resulted in a significant decrease in the proliferation rate of these primary tumor cells (*Figure 3I*), emphasizing the importance of Atmin function in supporting elevated *Pdgfra* expression in *p53$^{\Delta N}$*gliomas.

## Pdgfra overexpression is reversible with pharmacological Atm inhibition

Atmin has two known functions: an Atm-dependent function, in which Atmin interacts with Atm and is required for Atm signaling in several stress contexts, and an Atm-independent function, in which Atmin is required for transcription of the dynein light chain Dynll1. To determine whether acute silencing of *Atmin* in tumor cells might reduce *Pdgfra* expression by disrupting Atm signaling, we tested the effects of pharmacological Atm inhibition in *p53$^{\Delta N}$*primary tumor cells. Treatment with an

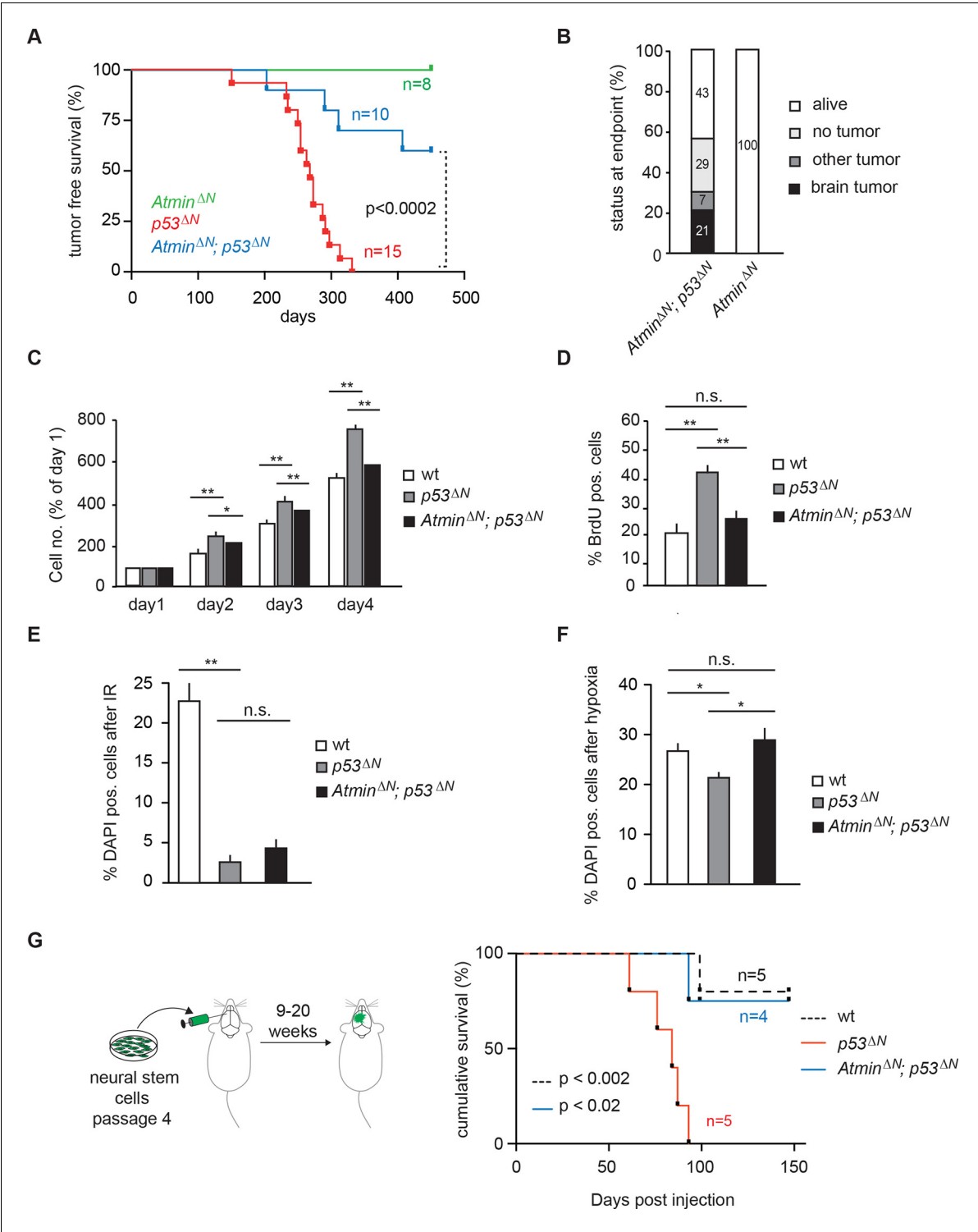

**Figure 2.** Loss of *Atmin* rescues GBM formation in *p53*$^{\Delta N}$ brains. (**A**) Kaplan-Meier curves showing tumor-free survival in *Atmin*$^{\Delta N}$ and *Atmin*$^{\Delta N}$; *p53*$^{\Delta N}$ mice. *p53*$^{\Delta N}$ curve from *Figure 1A* (same experiment) is shown for comparison. (**B**) Status of mouse cohorts at 450 days, showing tumor incidence. (**C**) Atmin loss rescues the increased proliferation of *p53*$^{\Delta N}$ NSCs. (**D**) Mean percentage of BrdU-positive NSCs from the indicated genotypes, assessed by FACS following a 2-hr BrdU pulse. (**E**) Mean number of DAPI-permeable (non-viable) cells after IR or (**F**) hypoxia, showing sensitivity of *Atmin*$^{\Delta N}$; *p53*$^{\Delta N}$ NSCs to hypoxia but not IR. n.s., not significant; * p<0.05, ** p<0.01. (**G**) Scheme of orthotopic NSC transplant experiment (left) and Kaplan-Meier curves indicating survival of NOD/SCID mice orthotopically transplanted with *p53*$^{\Delta N}$ and *Atmin*$^{\Delta N}$; *p53*$^{\Delta N}$ NSCs (right). Error bars represent the SEM of three biological repeats, and two biological repeats for (**E**). IR, Ionizing radiation; NSC, Neural stem cell.

*Figure 2 continued*

The following figure supplements are available for figure 2:

**Figure supplement 1.** Histological features of glioblastomas observed in *p53*<sup>ΔN</sup> and *Atmin*<sup>ΔN</sup>; *p53*<sup>ΔN</sup> animals.

**Figure supplement 2.** Examples of murine neural stem cells in culture.

**Figure supplement 3.** *Atmin*<sup>ΔN</sup> NSCs proliferate at a similar rate as wild-type NSCs.

**Figure supplement 4.** Gross genome stability is unaffected in *Atmin/Trp53* double mutant NSCs.

**Figure supplement 5.** Radiation-induced arrest is similarly impaired in *Trp53*-mutant and *Atmin/Trp53* double mutant NSCs.

**Figure supplement 6.** Tumors arising from orthotopic injection of NSCs are Nestin and Ki67-positive.

**Figure supplement 7.** Asymptomatic *Atmin*<sup>ΔN</sup>; *p53*<sup>ΔN</sup>-injected animals show persistent GFP-positive cells.

Atm inhibitor (ATMi) significantly reduced Pdgfra transcript and protein levels (*Figure 4A and B*) and efficiently reduced proliferation of *p53*<sup>ΔN</sup>primary tumor cells (*Figure 4C*), similar to our observations after *Pdgfra* silencing. ATMi treatment also reduced *Pdgfra* expression in *p53*<sup>ΔN</sup>NSCs (*Figure 4D* and *Figure 4—figure supplement 1*). Consistent with these results, *Atm*<sup>-/-</sup>; *p53*<sup>ΔN</sup>double-mutant NSCs had dramatically lower *Pdgfra* expression levels than *p53*<sup>ΔN</sup>NSCs, comparable to *Atmin*<sup>ΔN</sup>; *p53*<sup>ΔN</sup>and *wt* NSCs (*Figure 4E and F*). Furthermore, in vitro proliferation assays showed that genetic loss of *Atm* reduced the proliferation of *p53*<sup>ΔN</sup>NSCs to *wt* levels (*Figure 4G*). Importantly, the reduction in cell proliferation in vitro correlated with reduced tumorigenic potential of the NSCs when orthotopically injected into the brains of NOD/SCID mice. Mice injected with *Atm*<sup>-/-</sup>; *p53*<sup>ΔN</sup>NSCs survived up to 50% longer than those receiving *p53*<sup>ΔN</sup>NSCs (*Figure 4H*). However, *Atmin* deletion (*Figure 2G*) was more effective than deletion of *Atm* (*Figure 4H*). Atmin can also act as a transcription factor (*Jurado et al., 2012a*; *2012b*; *Goggolidou et al., 2014*), and it is conceivable that this Atm-independent function of Atmin contributes to the suppression on GBM. The similar effect of *Atmin* deletion and Atm inhibition on *Pdgfra* expression and proliferation of NSCs and primary tumor cells supports the hypothesis that Atmin functions in these processes via its modulation of Atm signaling. Given the extensively characterized role of PDGFRA in cell proliferation and in particular in glioma, it is reasonable to propose that the reduction in *Pdgfra* expression upon *Atm* loss contributes to the reduced tumorigenic potential observed.

Since Atm is best known as a DNA damage signaling kinase, we examined whether the effect of Atm on *Pdgfra* expression and tumorigenicity was related to altered DNA damage signaling in *p53*<sup>ΔN</sup>glioma cells. Neither *p53*<sup>ΔN</sup>nor *Atmin*<sup>ΔN</sup>; *p53*<sup>ΔN</sup>NSCs showed increased γh2ax or 53bp1 foci in basal conditions (*Figure 5A and B*), suggesting that the different proliferation rates are not caused by changes in endogenous DNA damage. *Pdgfra* expression was not affected by IR or hydroxyurea (HU), two well-described inducers of ATM signaling, either in wt or in *p53*<sup>ΔN</sup>NSCs (*Figure 5C*) and no increased Atm substrate phosphorylation could be detected in untreated *p53*<sup>ΔN</sup> NSCs (*Figure 5D*), suggesting that *Pdgfra* upregulation is not a consequence of stimulation of DNA damage signaling. Atm substrate phosphorylation was also comparable in IR- and HU-treated *wt*, *p53*<sup>ΔN</sup>and *Atmin*<sup>ΔN</sup>; *p53*<sup>ΔN</sup>NSCs (*Figure 5D*), suggesting that the response to DNA damage stimuli is comparable in these cells. This suggests that while Atm is required for the increased *Pdgfra* expression in *p53*<sup>ΔN</sup> cells, this does not involve alterations in IR or HU-induced Atm signaling, but at this point does not exclude the possibility that increased ROS (reactive oxygen species) levels in *p53*<sup>ΔN</sup> GBM cells might contribute to Atm pathway activation and subsequent *Pdgfra* induction and GBM development.

## Primary human glioma neural stem cells are sensitive to ATM inhibition and combinatorial treatment with PDGFR inhibitors

To investigate the relevance of ATM signaling for *PDGFRA* upregulation in human GBM, we took advantage of patient derived human GBM neural stem cells (GNSCs) (*Stricker et al., 2013*) that

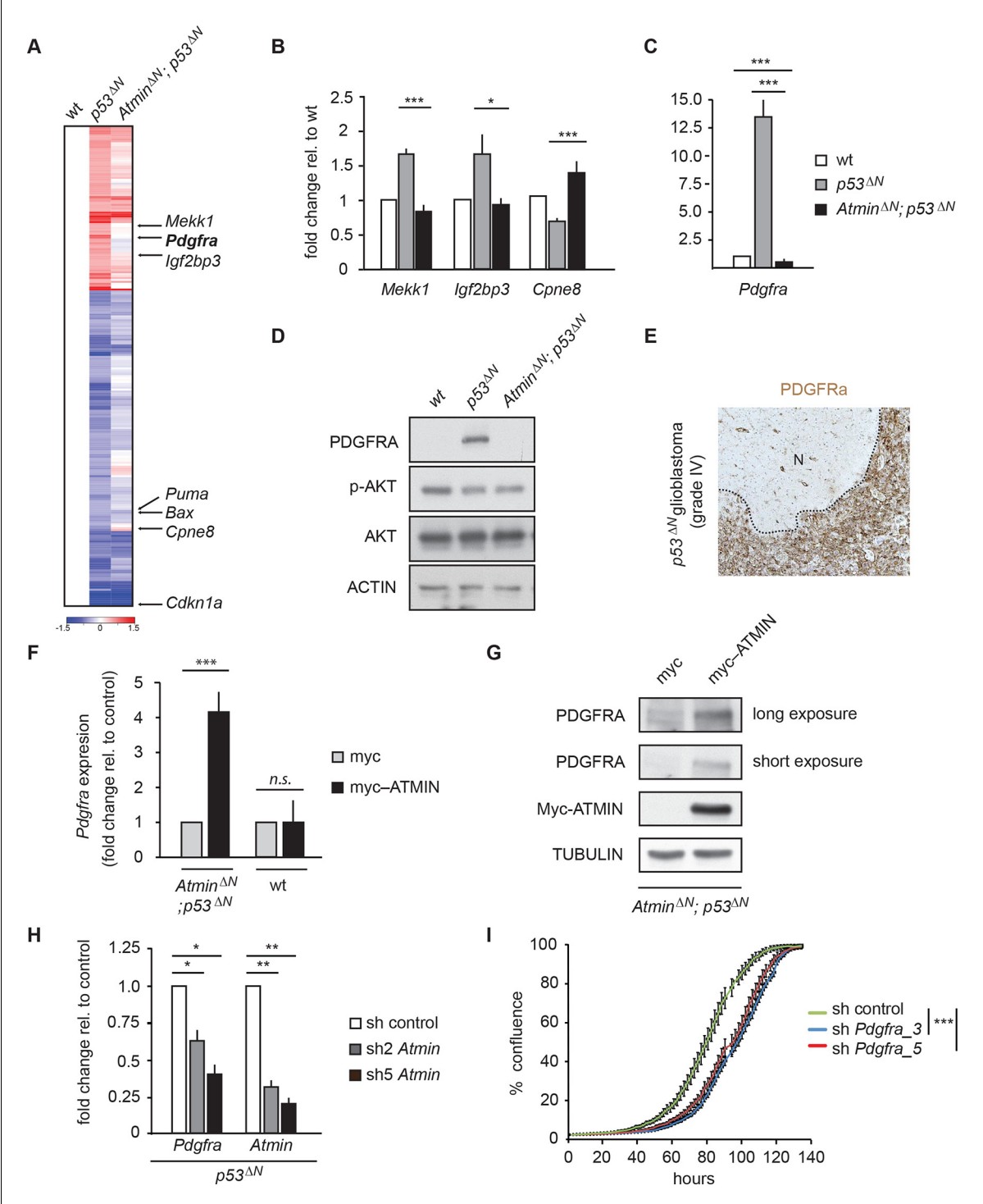

**Figure 3.** Atmin regulates *Pdgfra* expression in a mutant *Trp53* background. (**A**) Heatmap showing genes deregulated more than 1.5-fold in *Trp53* null NSCs and their corresponding expression in *Atmin*ΔN; *p53*ΔN double null NSCs. (**B**, **C**) qRT-PCR validation of GBM implicated genes that show deregulated expression in *p53*ΔN but not *Atmin*ΔN; *p53*ΔN NSCs. (**D**) Elevated Pdgfra protein expression in *p53*ΔN but not *Atmin*ΔN; *p53*ΔN NSCs. Corresponding p-Akt and total Akt levels were probed to analyze pathway activation. Actin was used as loading control. (**E**) Tumor section from a GBM arising from a *p53*ΔN animal showing elevated Pdgfra expression. 'N' denotes the normal brain region adjacent to the tumor mass. Dotted lines indicate tumor border. (**F**) Myc-Atmin overexpression increases expression of *Pdgfra* mRNA in *Atmin*ΔN; *p53*ΔN but not wt cells. (**G**) Myc-Atmin overexpression increases Pdgfra protein levels in *Atmin*ΔN; *p53*ΔN NSCs. Tubulin was used as loading control. (**H**) *Atmin* silencing using two independent shRNA constructs reduces *Pdgfra* expression, assessed by qRT-PCR. (**I**) Stable *Pdgfra* silencing using two independent shRNA constructs

*Figure 3 continued on next page*

*Figure 3 continued*

reduces primary tumor cell proliferation, measured using IncuCyte timelapse microscopy. Error bars represent the SEM of at least three biological repeats and two biological repeats in (I).* p<0.05, ** p<0.01, *** p<0.001. NSC, Neural stem cell

The following figure supplements are available for figure 3:

**Figure supplement 1.** The transcriptional profile of $p53^{\Delta N}$ NSCs is most closely related to the human proneural GBM subtype, relative to expected distances.

**Figure supplement 2.** Pdgfra expression is elevated in $p53^{\Delta N}$ tumors, but Egfr is not.

**Figure supplement 3.** Some markers frequently altered in GBM are unaltered in $p53^{\Delta N}$ NSCs.

**Figure supplement 4.** Pdgf ligand expression is not significantly altered in $p53^{\Delta N}$ NSCs.

**Figure supplement 5.** Two independent shRNAs cause *Pdgfra* knockdown.

**Figure supplement 6.** miR34a expression is reduced in $p53^{\Delta N}$ NSCs.

---

have documented *TP53* mutations (lines G26, G166, and G179, *Figure 6A*). We confirmed that cell extracts from each of these human tumors were unable to activate a luciferase construct containing multiple TP53-binding elements (*Figure 6—figure supplement 1*). *PDGFRA* expression was significantly increased in two out of three human GNSC lines compared with untransformed human neural stem cells (cb660; (*Engström et al., 2012*; *Pollard et al., 2009*) (*Figure 6B*). To assess whether this observation is representative of the wider human GBM spectrum, we analyzed the gene expression patterns in the 'Glioblastoma Multiforme (TCGA, provisional)' dataset from The Cancer Genome Atlas. We analyzed 153 RNA-seq and 500 microarray datasets for 518 glioblastoma samples and completed these with the clinical subgroup information from (*Brennan et al., 2013*). The gene expression datasets contained z-scores representing the differences in expression levels of *PDGFRA* between cancer and control samples (*Figure 6—figure supplements 2* and *3*). The RNA-seq and microarray data were handled separately; for each, we observed higher average *PDGFRA* expression in *TP53*-mutant glioblastoma samples compared with *TP53* wild-type samples (*Figure 6C,D*). We also observed higher *PDGFRA* expression levels in the Proneural and GCiMP GBM subtypes when compared to the Classical, Neural, and Mesenchymal (*Figure 6E* and *Figure 6—figure supplement 4*).

In line with the data from our mouse model, treatment with an ATMi strongly reduced *PDGFRA* expression in GNSCs, whereas no significant effect on *PDGFRA* expression was detected in cb660 controls (*Figure 6F*). To determine whether a reduction in *PDGFRA* levels would be sufficient to inhibit the proliferation of primary GNSCs from human tumors, G179 cells were infected with doxycycline (dox)-inducible shRNA constructs targeting *PDGFRA* (*Figure 6G* and *Figure 6—figure supplement 5*), and their proliferation was monitored over a period of 10 days. In agreement with our results in the mouse, depletion of *PDGFRA* led to a significant decrease in the proliferation rate of human GNSCs (*Figure 6H*). Thus, both murine and human TP53-mutant glioma cells are sensitive to loss of PDGFRA.

To analyze the physiological effects of ATM inhibition on GNSCs, proliferation was monitored over a period of 7 days. Similar to our observations in murine NSCs, ATMi treatment reduced the proliferation rate of human TP53 mutant GNSCs, but not cb660 control cells (*Figure 7A*). PDGFR inhibitors (PDGFRi) have previously been shown in vitro and in vivo to reduce tumor growth in a glioma model (*Kilic et al., 2000*). These advances, however, have so far failed to transition into clinical practice (*Wen et al., 2006*; *Rich et al., 2004*). We reasoned that ATMi-mediated reduction in PDGFRA protein levels could potentiate the effects of PDGFR inhibition. Consistent with this, we observed that co-treatment of GNSCs, but not untransformed NSCs (e.g. cb660), with relevant ATM and PDGFR inhibitor concentrations further reduced cell proliferation (*Figure 7A* and *Figure 7—figure supplement 1*). More detailed analysis revealed that this treatment combination induced substantial cell death, while untransformed NSCs remained largely unaffected (*Figure 7B,C*). The

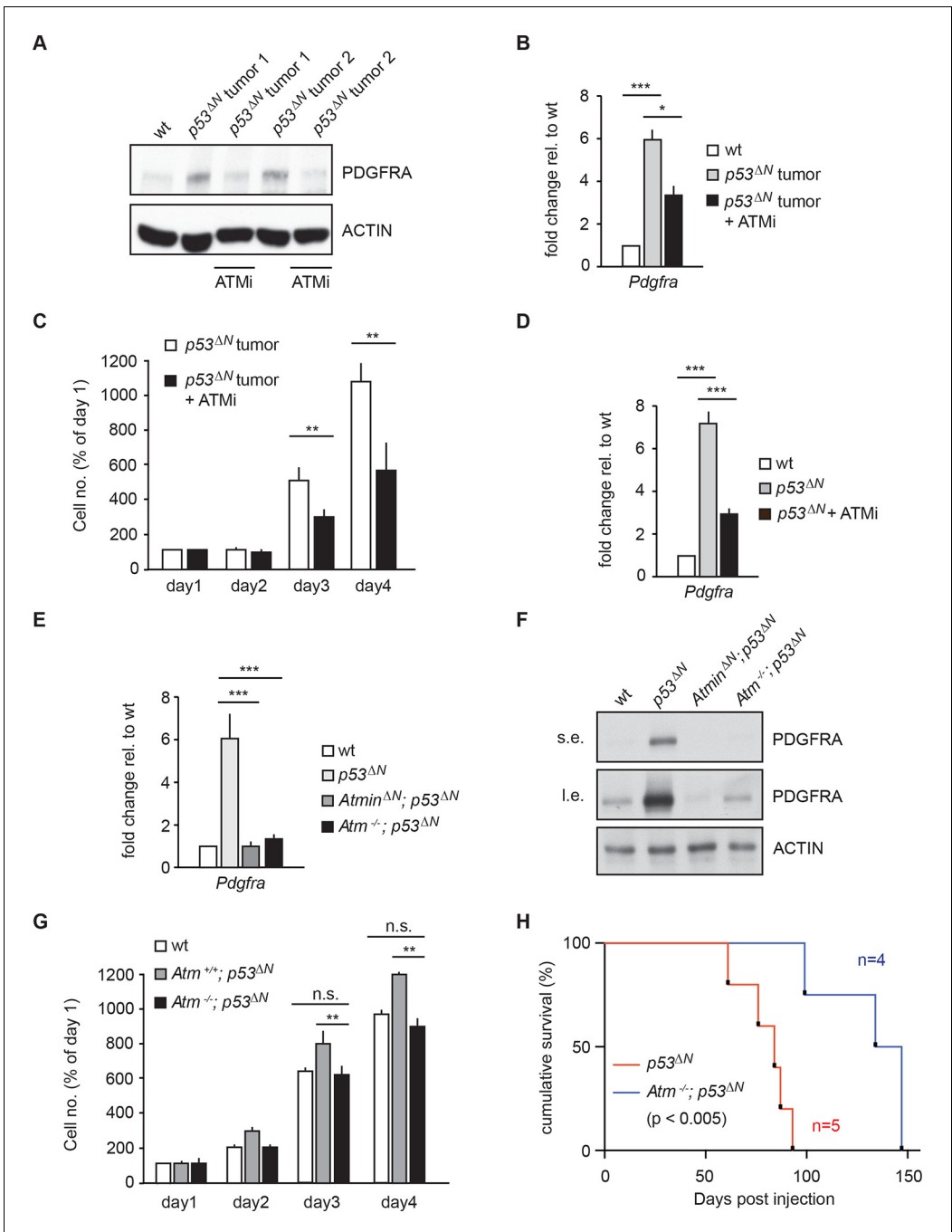

**Figure 4.** Atm inhibition reduces *Pdgfra* expression and reduces tumorigenic potential in murine *p53*^ΔN primary tumor cells and NSCs. (A–B) Atm inhibitor (ATMi) treatment reduces Pdgfra protein levels (A) and *Pdgfra* expression (B) in *p53*^ΔN primary tumor cells. (C) ATMi treatment reduces in vitro proliferation of *p53*^ΔN primary tumor cells. Error bars represent the SEM of two biological repeats. (D) ATMi reduces *Pdgfra* expression in *p53*^ΔN NSCs, assayed by qPCR. (E–F) *Pdgfra* expression (E) and Pdgfra protein levels (F) are reduced in *Atm*^-/-; *p53*^ΔN and *Atmin*^ΔN; *p53*^ΔN compared with *p53*^ΔN NSCs. cDNA is normalized to *Actin* levels. Error bars represent the SEM of at least three biological repeats. s.e. and l.e. denote short and long exposures of the same blot, respectively. (G) Genetic loss of *Atm* reduces in vitro proliferation of *p53*^ΔN NSCs. Error bars represent the SEM of two biological repeats. (H) Kaplan-Meier curves indicating increased survival of NOD/SCID mice orthotopically transplanted with *Atm*^-/-; *p53*^ΔN NSCs compared with *p53*^ΔN NSCs. n.s., not significant; * p<0.05, ** p<0.01, *** p<0.001. NSC, Neural stem cell.

The following figure supplement is available for figure 4:

**Figure supplement 1.** ATM inhibitor (ATMi) treatment reduces Pdgfra protein levels in *p53*^ΔN NSCs. NSCs, Neural stem cells.

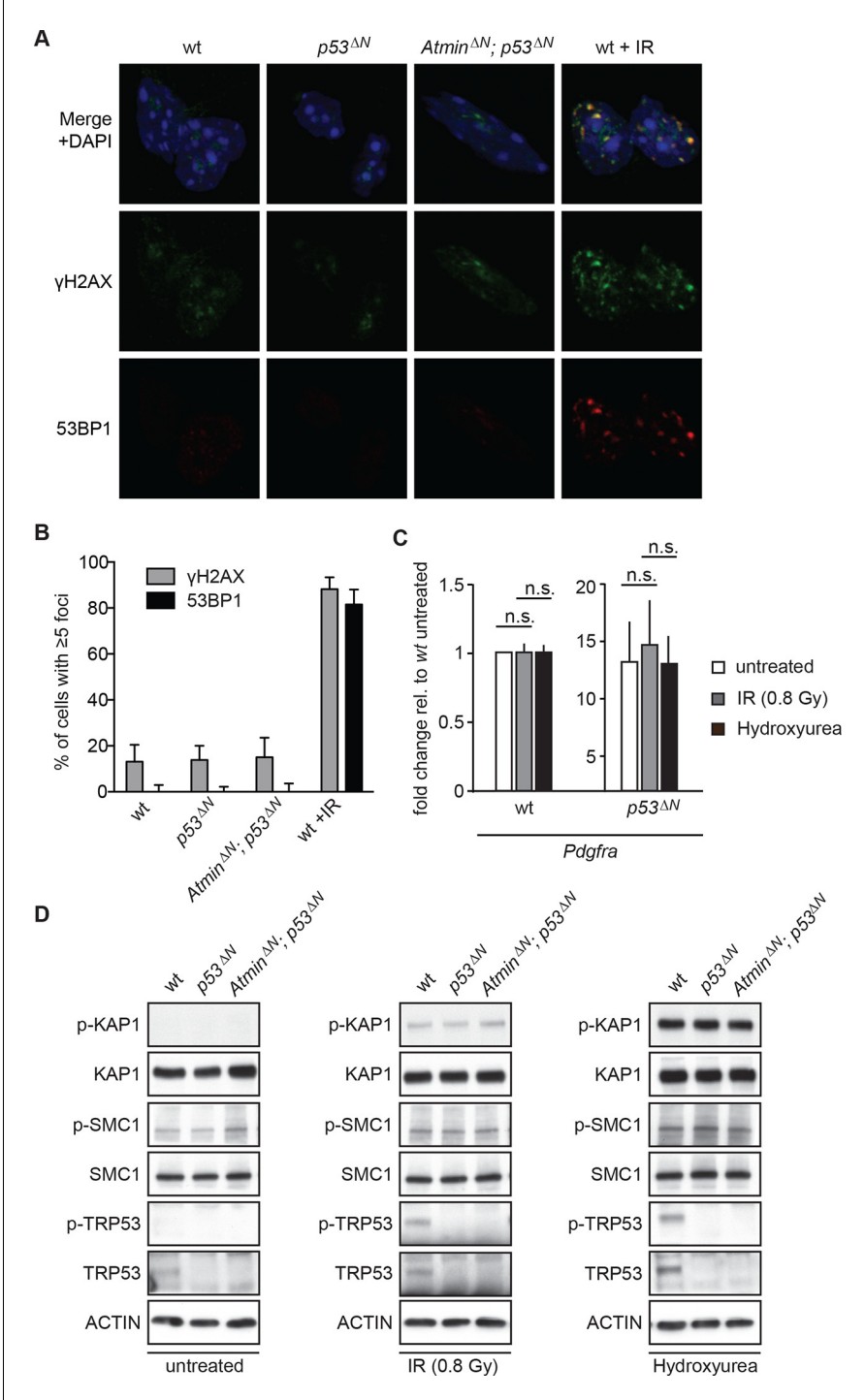

**Figure 5.** Atm signaling in response to DNA damaging agents is remarkably unaffected in *p53*ᐃN and *Atmin*ᐃN; *p53*ᐃN cells and DNA damaging treatments do not affect *Pdgfra* expression. (**A**) *p53*ᐃN and *Atmin*ᐃN; *p53*ᐃN NSCs do not show elevated γh2ax and 53bp1 foci in untreated conditions, suggesting low endogenous damage. Cells treated with 5Gy IR (wt +IR) are shown as a positive control. (**B**) Quantification of γh2ax and 53bp1 foci in (**A**). Error bars indicate 95% confidence intervals. (**C**) qPCR of *Pdgfra* in wt and *p53*ᐃN NSCs in untreated conditions and after DNA damage-inducing stimuli. cDNA is normalized to *Actin* levels. Error bars represent SEM of three biological repeats. (**D**) Western blots depicting Atm substrate phosphorylation after the indicated stimuli in NSCs of different genotypes.

Cancer biology

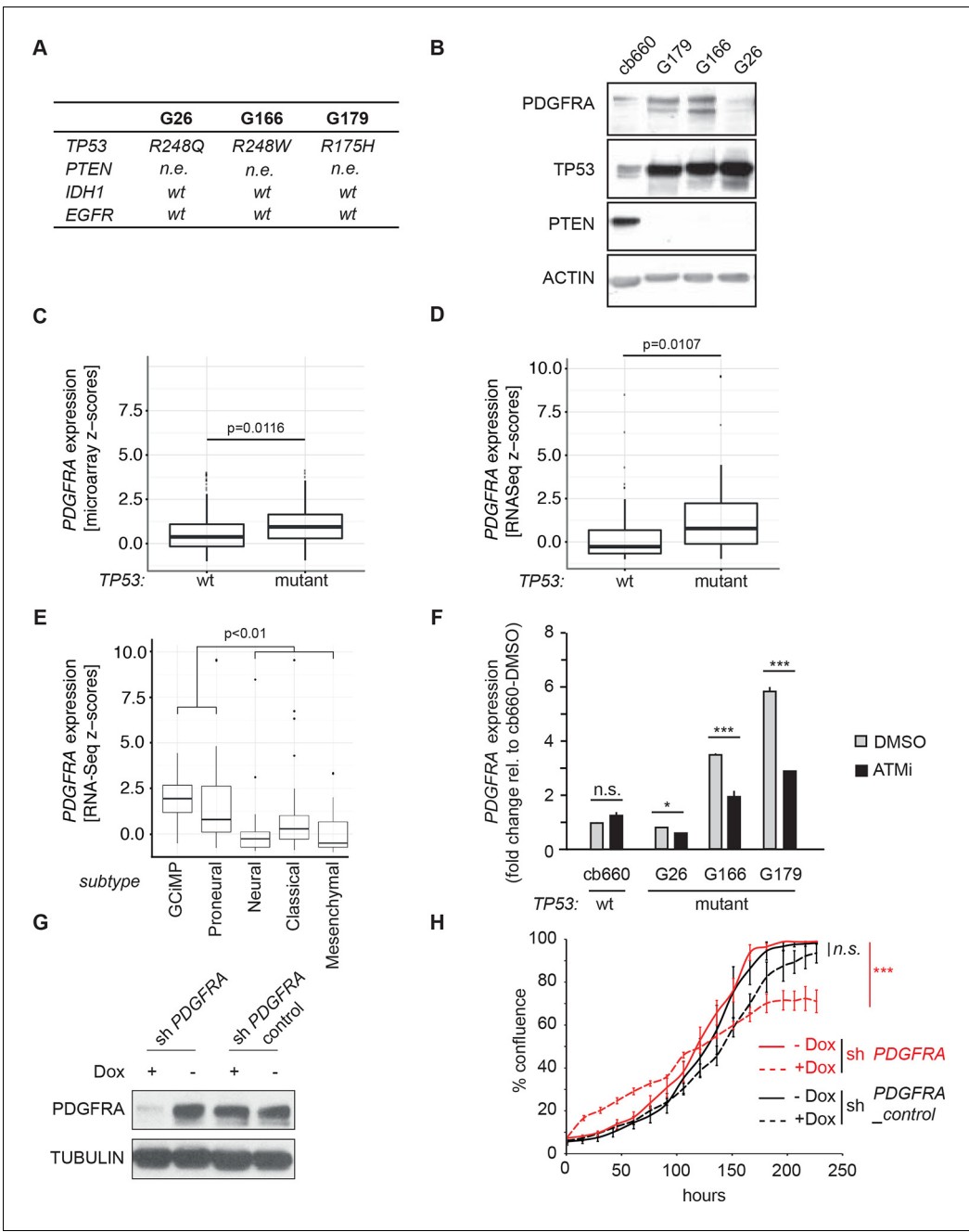

**Figure 6.** ATM inhibitor treatment reduces elevated PDGFRA expression and decreases proliferation in human GBM tumor cells. (**A**) Mutational status of *TP53*, *PTEN*, *IDH1*, and *EGFR* in the indicated glioma neural stem stell (GNSC) lines. n.e., not expressed. wt, wild type. (**B**) Western blot depicting PDGFRA, TP53, PTEN, and ACTIN expression in three GNSC lines (G26, G166, and G179) and cb660 control cells. (**C,D**) Box plots showing *PDGFRA* expression in TP53-wt and TP53-mutant TCGA human glioblastoma datasets, measured by microarray (n=153). (**C**) or RNA-Seq (n=500). (**D**) Data from 518 patient samples in total with an overlap of 135 patients. (**E**) Box plots showing *PDGFRA* expression z-scores in TCGA human glioblastoma subtypes, measured by RNA-Seq (n=150). p-Values in (**C–E**) calculated using Wilcoxon's test. (**F**) qRT-PCR showing decreased *PDGFRA* expression in TP53-mutant GNSC lines after ATM inhibitor (ATMi) treatment. cDNA is normalized to GAPDH levels. Error bars represent the STDEV of two biological repeats. (**G**) Western blot showing knockdown of PDGFRA using a doxycycline (Dox)-inducible shRNA construct. (**H**) PDGFRA knockdown using doxycycline (Dox)-inducible shRNA reduces proliferation of TP53-mutant G179 GNSCs. Cell confluence measured by IncuCyte timelapse microscopy. Error bars represent the SEM of three biological repeats. * p<0.05, *** p<0.001, n.s., not significant.

*Figure 6 continued on next page*

*Figure 6 continued*

The following figure supplements are available for figure 6:

**Figure supplement 1.** Luciferase reporter assay for TP53 activity using the p53-550RE construct in human GNSC (G26, G166, G179) and control NSC (cb660) lines. NSC, Neural stem cell.

**Figure supplement 2.** Correlation of *PDGFRA* expression levels in TCGA glioblastoma samples represented in both microarray and RNASeq datasets.

**Figure supplement 3.** Comparison of the distribution of *PDGFRA* expression levels in human glioblastoma samples measured by RNASeq and by microarray.

**Figure supplement 4.** Box plots showing *PDGFRA* expression z-scores in TCGA human glioblastoma subtypes measured by microarray (n=487).

**Figure supplement 5.** Control for *Figure 6G and H* showing efficient knockdown of *PDGFRA* after doxycycline administration.

---

proportion of apoptotic GNSCs receiving the combination treatment was quantifiable by microscopic detection of a caspase-cleaved fluorescent substrate after up to four days of drug treatment; thereafter, dead cells detached from the plate, precluding accurate quantification. Essentially, all GNSCs were eventually killed by the combination of ATM and PDGFR inhibitors, as judged by cell detachment. In contrast, PDGFRi treatment by itself had little effect on cell death and proliferation (*Figure 7 A–C*). Seeding GNSCs at higher confluence gave similar results (*Figure 7—figure supplement 2*), as did a second, independent PDGFRi in combination with the ATMi (*Figure 7—figure supplement 3*), indicating that cell death was not an indirect consequence of slower proliferation induced by the ATMi or a side effect of a particular drug. Hence, ATM is required for *PDGFRA* overexpression in both murine and human GBM cells, and combined inhibition of ATM and PDGFR induces lethality in TP53-mutant glioma cells, promising new opportunities for future GBM treatment.

Our data indicate that disruption of Atm/Atmin function not only inhibits GBM initiation in Trp53-deficient animals, but also reduces the tumorigenic potential of established human glioma cells, suggesting that ATM inhibitors might be valuable tools in GBM therapy.

## Discussion

An important goal of personalized cancer medicine is to identify vulnerabilities of specific tumor genotypes. Since TP53 signaling is the most common genetically altered pathway in human gliomas (*Cancer Genome Atlas Research Network, 2008*), it is desirable for future cancer therapies to identify new molecular targets that affect TP53-deficient cells. Here, we have shown that genetic loss of *Atmin* or *Atm* reduces GBM formation initiated by deletion of *Trp53*. This finding indicates that Atm/Atmin is crucial for Trp53-deficient GBM development, and suggests the use of ATM inhibitors, which have already been developed, for GBM therapy (*Basu et al., 2012*; *Golding et al., 2009*; *2012*; *Batey et al., 2013*).

Although other groups have previously observed that loss of *Trp53* is sufficient to provoke GBM formation (*Wang et al., 2009*; *Zheng et al., 2008*), the potential advantages of this model system have largely been overlooked. Even on a mixed genetic background, mice harboring a Nestin-driven Trp53 deletion show strong consistency in tumor latency, high penetrance of brain tumors and high-grade GBM formation. Such a model, incorporating the stochastic evolution and genetic heterogeneity of human glioblastoma patients, complements existing genetically defined mouse models that show an accelerated disease course (*Chen et al., 2012*).

Atm is a rare example of a protein required for formation of a *Trp53*-deficient tumor that is amenable to pharmacological inhibition, with potential direct therapeutic implications. This pro-tumorigenic function for Atmin and Atm signaling in brain cancer is unexpected, as systemic Atm deletion has previously been found to accelerate Pdgf ligand-induced gliomagenesis (*Squatrito et al., 2010*). However, the previous study was carried out in Trp53 wild-type animals. It was recently shown that

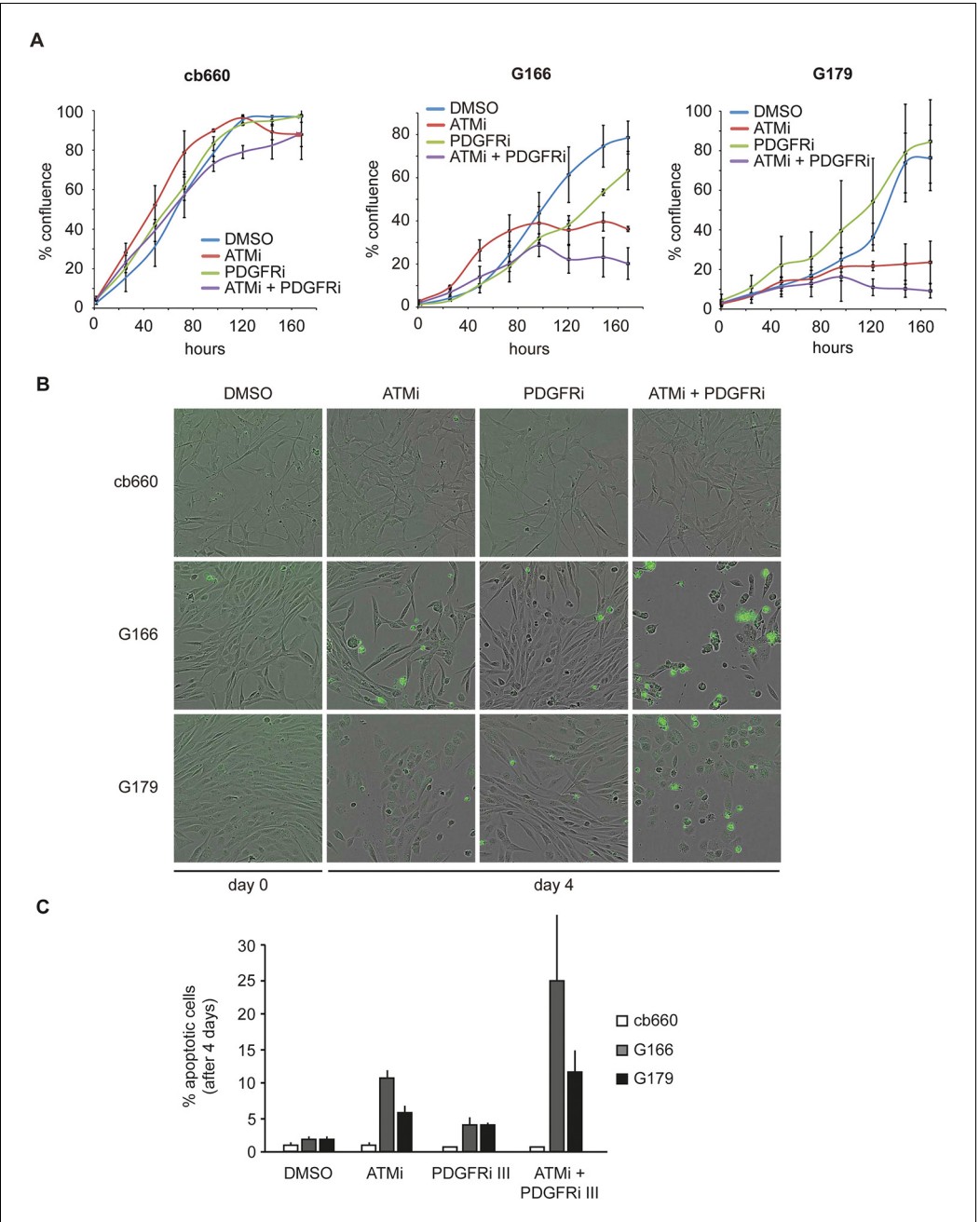

**Figure 7.** Combinatorial treatment with both ATM and PDGFR inhibitors induces apoptosis of human glioma stem cells. (**A**) Growth curves showing the proliferation of wild-type control human neural stem cells (cb660) and two independent cell lines isolated from TP53-deficient human gliomas (G166 & G179) over 7 days in the presence of vehicle control (DMSO), ATM inhibitor (ATMi), PDGFR inhibitor (PDGFRi), or both inhibitors. Error bars depict STDEV from two biological repeats. (**B**) Representative bright-field/fluorescent images of the same cell types treated as above after 4 days. Cells undergoing apoptosis are labeled by emitting GFP. (**C**) Quantification of apoptotic cells after the treatments indicated above, represented as % apoptotic cells from total live cells. Error bars depict STDEV from three biological repeats. After four days, it became impossible to quantify apoptotic GNSCs receiving the combination treatment, since they detached completely from the culture plate. These percentages thus represent an underestimate of the total extent of G166 and G179 cell death with ATMi + PDGFRi.

The following figure supplements are available for figure 7:

*Figure 7 continued on next page*

*Figure 7 continued*

**Figure supplement 1.** ATM and PDGFR inhibitors were used at minimally required concentration to achieve pathway inhibition.

**Figure supplement 2.** Bright-field images of timelapse microscopy using cells seeded at high confluency.

**Figure supplement 3.** Apoptosis quantification of a second PDGFR inhibitor.

TP53 status is crucial in determining the cellular response to ATM inhibition in human glioma cell lines (*Biddlestone-Thorpe et al., 2013*), so this difference might explain the apparently opposite function of Atm in our model. TCGA data indicates that ATM mutation/deletion is rare in TP53-mutant GBM, supporting the hypothesis that ATM loss is not a tumor driver and may even inhibit tumorigenesis in this context. Because the pleiotropic effects of ATM signaling are highly context dependent, factors other than TP53 status must also influence the outcome in vivo. For example, loss of Atm cooperates with Trp53 deficiency and accelerates the progression of T-cell lymphoma (*Westphal et al., 1997*). Thus, the relationship between TP53 and ATM signaling in tumorigenesis may be tissue specific, ranging from cooperation to antagonism.

Adding to this context dependency, ATM's function in the DNA damage response inhibits tumor-igenesis in precancerous lesions but can also promote resistance to DNA damaging therapies in established tumors. Indeed, recently published work has shown that ATM inhibition preferentially radiosensitizes TP53 mutant GBM cell lines (*Biddlestone-Thorpe et al., 2013*). Our data reveal an additional pro-tumorigenic effect of Atm, which acts in the absence of ionizing radiation and involves non-canonical, Atmin-dependent signaling. It is possible that the effect of Atm inhibition in our model would also be enhanced by ionizing radiation, since the radiosensitizing effect would be predicted to be largely independent of Atmin (*Kanu and Behrens, 2007*). Thus, combination treatment with ATM and PDGFR inhibitors together with radiotherapy may represent a therapeutic opportunity for GBM.

Our results indicate that the requirement for the Atm/Atmin pathway in $p53^{\Delta N}$ GBM lies in the initiation of a program of altered gene expression (*Figure 3A*) and does not directly involve the DNA damage response to double-strand breaks induced by IR or HU (*Figure 5B and C*). Although Atmin has been implicated directly as a transcription factor for *Dynll1* (*Jurado et al., 2012b*), a broader role for Atmin in transcriptional regulation had not yet been determined. We find here that *Atmin* loss leads to widespread rescue of glioma-associated changes in gene expression in the $p53^{\Delta N}$ background. Several of these genes could be collectively responsible for suppressing GBM formation in $Atmin^{\Delta N}$; $p53^{\Delta N}$ animals. The most compelling single candidate, however, is the proto-oncogene *Pdgfra*. It has been shown that stimulating Pdgf signaling in vivo is sufficient to induce glioma-like growths (*Jackson et al., 2006*). Moreover, *PDGFRA* is commonly overexpressed in human glioma (*Engström et al., 2012*), but only a subset of gliomas display *PDGFRA* locus amplification (*Furnari et al., 2007*) (*Cancer Genome Atlas Research Network, 2008*), indicating that additional mechanisms drive its increased expression. *PDGFRA* overexpression frequently correlates with LOH on chromosome 17p, where the human *TP53* gene is located (*Hermanson et al., 1996*), supporting the view that the oncogenic misexpression of *PDGFRA* might be a direct consequence of TP53 inactivation in human GBM. In our murine and human glioma lines, Pdgfra levels could be dynamically modulated by depleting Atmin or inhibiting Atm. At present, how Atm/Atmin controls Pdgfra expression in this background is unclear. Chromatin immunoprecipitation experiments in murine NSCs indicated that Atmin does not bind the *Pdgfra* gene (unpublished observations), making the regulation unlikely to be via direct transactivation, most likely involving currently unidentified intermediate transcriptional regulators of *Pdgfra* expression. We speculate that Trp53 controls the expression of a negative regulator of Pdgfra, for example, a repressor or a miR RNA, and that Atm signalling is required for the expression or function of this factor. As these intermediate factors could be promising targets for pharmacological inhibition, further investigation into their identity is of therapeutic interest.

Although the oncogenic role of PDGFR signaling in glioma is well established, the clinical efficacy of PDGFR inhibitors has so far been disappointing (*Wen et al., 2006*; *Rich et al., 2004*). Here, we

demonstrate that the effect of these drugs on the proliferation and survival of human glioma cells can be greatly potentiated by combining PDGFR inhibitor treatment with ATM inhibitors. Importantly, the strongest effects of combinatorial treatment on cell proliferation and survival are specific to tumor cells that have lost TP53 function, suggesting that systemic therapy should target TP53-deficient tumors with minimal impairment to normal tissue.

A second encouraging result of combining ATM and PDGFRA inhibitors is that they not only reduce tumor cell proliferation, but also potently induce cell death. While our data support cooperation based on simultaneous inhibition of PDGFR signaling at the transcriptional and protein kinase levels, it is likely that this synergy also encompasses broader effects of ATM and PDGFRA inhibition. In addition, current PDGFR inhibitors are not completely specific and also inhibit EGFR, VEGFR, FLT3, and c-KIT kinases amongst others (*Andrae et al., 2008*; *Homsi and Daud, 2007*). Similarly, ATM inhibitors have also been shown to display broader kinase inhibitor activity, especially at higher concentrations (*Hickson et al., 2004*). Thus, PDGFR-independent mechanisms could contribute to the observed treatment synergy.

In summary, we have identified a novel protumorigenic function of ATM signaling in GBM. Our results present a rationale for expanding the investigation of ATM inhibitors from radiosensitizers to potential therapies in their own right, and point toward improvements in the efficacy of PDGFRA inhibition using combination treatment.

## Materials and methods

### Ethics statement

Experiments in mice were carried out with the approval of the Crick Institute's Ethical Review Committee and under the guidance of the Biological Resources Unit.

### Animal models

*Atmin^{f/f}*, *p53^{f/f}*, *Atm^{-/-}*, and *Nestin-Cre* mice have been described previously (*Kanu et al., 2010*; *Jonkers et al., 2001*; *Barlow et al., 1996*; *Tronche et al., 1999*). Immunocompromised NOD/SCID mice were maintained in-house. Mice were maintained and bred on a mixed background in pathogen-free conditions, monitored for signs of ill health and culled when moribund. Strain background had no significant effect on the latency or development of gliomas in *p53^{ΔN}* mice.

### Cell culture

NSCs were isolated as spheres from fore and midbrains of mouse E13.5 embryos. Cells were initially cultured as spheres under self-renewal conditions, as previously described. Adherent NSC cultures were derived as previously described (*Pollard et al., 2006*; *Conti et al., 2005*) with minor modifications (*Sancho et al., 2013*). Briefly, primary spheres were plated in Neurobasal Medium (Invitrogen, Grand Island, NY, USA) supplemented with 1% Penicillin/Streptomycin (Invitrogen), 1% L-glutamine (Invitrogen), 2% B27 supplement (Invitrogen), 1% N-2 supplement (Invitrogen), 20 ng/ml EGF (PeproTech), 20 ng/ml FGF-basic (PeproTech), and 1 µg/ml laminin (Sigma). All experiments were performed using undifferentiated adherent NSCs (see *Figure 2—figure supplement 2*).

Primary tumor cells were generated from brain tumor samples of symptomatic mice (*Figure 3—figure supplement 5*). Tissue was subjected to mechanical and enzymatic dissociation, and single cells initially cultured to form spheres. Tumor spheres formed were then maintained as adherent tumor cell cultures in NSC media as described above.

### Plasmids and reagents

pCMV6-myc-ATMIN was generated by cloning mouse *Atmin* into a pCMV6 backbone. Silencing and mismatch constructs for *Atmin* have been described previously (*Kanu and Behrens, 2007*). The luciferase p53 response element (pGL3-550RE) was kindly provided by Karen Vousden. The ATM inhibitor Ku55993 (Merck) was used at 10 µM and replenished every 24 hr.

For DNA damage induction, NSCs of the indicated genotypes were either left untreated or subjected to 0.8 Gy irradiation or 2 mM hydroxyurea (HU) and harvested after 1 hr and 3 hr respectively for RNA extraction or protein lysates.

## Transfection and luciferase assay

For transfection, NSCs and GNSCs were plated at subconfluence and transfected with Lipofectamine 2000 according to the manufacturer's protocol (Invitrogen).

For luciferase assays, samples were transiently transfected with Firefly and Renilla luciferase reporters and luciferase activity was measured using the Dual-Luciferase Reporter Assay System (Promega), 36-hr post-transfection. Data are expressed as fold induction of luciferase activity after being normalized to expression of thymidine kinase-renilla luciferase (TK-renilla).

## Western blots and immunoprecipitation

For Western Blots (WB), NSCs were extracted in RIPA lysis buffer (NEB) supplemented with protease inhibitors (Sigma). For immunoprecipitation (IP) cells were lysed in IP buffer (20mM sodium phosphate buffer, 1 mM EDTA, 0.2% NP40, 150 mM NaCl supplemented with Na-orthovanadate, PMSF, NaF and protease inhibitor mixture [Sigma]). After sonication and centrifugation, supernatant was incubated overnight at 4°C with PDGFRA antibody, followed by 2-hr incubation with Dynabeads M-280 (Invitrogen), washed with IP buffer (0 mM NaCl, 150 mM NaCl and 1 M NaCl) and eluted in Laemmli sample buffer. All primary antibodies were used at 1:1000 dilution and secondary antibodies at 1:10000. The following antibodies were used: p53, Pdgfra, p-Pdgfra, p-Chk2, p-Akt, Akt (all Cell Signaling); Myc-9E10 (CRUK); Smc1 (Abcam); p-Smc1, Chk2, Tubulin (all Merck); p-Kap1, Kap1 (both Bethyl Labs); Atm (Santa Cruz), p-Atm (Epitomics), p-p53, Actin, GAPDH, HRP-conjugated goat anti-mouse/rabbit IgG (all Sigma).

## Quantitative RT-PCR

For qRT-PCR analysis, mRNA was isolated from NSCs using the RNeasy mini kit (Qiagen). Results (normalized to *Actin* expression) are presented as fold change relative to control. Sequences of primers used are listed in *Supplementary file 2*.

## Cell proliferation and hypoxia assays

For cell proliferation assays, NSCs were plated in duplicate in 12-well plates and cell number measured every day using a ViCell cell counter. The initial number of cells seeded was used to normalize to 100%. Mean averages were taken from at least four independent lines per genotype taken from different embryos and crosses. For BrdU profiling, cells pulsed with BrdU for 2 hr were fixed in 70% ethanol, stained with BrdU antibody and propidium iodide (PI) and analyzed by flow cytometry.

For hypoxia assay, NSCs were seeded in duplicate into 6-well plates and next day placed into a hypoxic chamber at 0.1% $O_2$ for 72 hr. ATM inhibitor was replenished after 24 hr and 48 hr. For cell death analysis, control-treated and hypoxia-treated cells were subjected to DAPI staining and analyzed via flow cytometry. Mean averages were taken from at least three independent experiments.

## IR-induced death and arrest

To determine IR-induced death, NSCs were irradiated at 5Gy and after 24 hr incubated with 4'-6-Diamidino-2-phenylindole (DAPI; Sigma) and assessed for apoptosis via flow cytometry. G1/S phase arrest was assessed by irradiating NSCs at 5Gy and after 18 hr pulsing with BrdU for 1.5 hr. Cells were prepared for flow cytometry as described above and gated on BrdU-positive cells. Mean averages were taken from at least three independent experiments.

## Orthotopic transplants

$1 \times 10^5$ early passage (p.4) GFP labeled NSCs of the indicated genotypes were injected using a stereotaxic frame into the striatum of 6- to 8-week-old mice (NOD/SCID strain), following administration of general anaesthesia as previously described. Animals were monitored daily and culled when moribund.

## Human NS and GNS cell culture

Fetal NS cell lines and GNS lines derived from human glioma samples have been described previously (*Stricker et al., 2013*; *Pollard et al., 2009*). The same culture conditions apply as previously described for mouse NSCs. Cell proliferation was measured over a period of four or seven days using an incubator microscope system for live cell imaging and measurement of cell confluence over

several days (IncuCyte) and cell death was quantified using CellPlayer Caspase 3/7 Reagent from Essen Bioscience. Cells were seeded at $1.2x10^4$ or $2.4x10^4$ into a 48-well plate and images taken every 30 min. The ATM inhibitor KU55993 was added at 10 µM. PDGFR inhibitors III and V (CAS 205254-94-0 and CAS 347155-76-4 respectively, both Merck) were used at 1µM and replenished every 24 hr over a period of four or seven days as indicated. Stable sh*PDGFRA* GNSC lines were generated by viral infection of GNSCs with pTRIPZ-sh*PDGFRA* targeting and control vectors (Thermo Fisher). Doxycycline was added at a final concentration of 0.5 µg/ml and replenished every day.

## Histological analysis and tumor grading

All mice were euthanized in a $CO_2$ chamber to preserve brain tissue. Brain sections were cut at 4 µm for Hematoxylin & Eosin (H&E) staining and all antibody staining. The following antibodies were used for IHC: Nestin (BD Bioscience), Gfap (Dako), Olig2, Neun (both Merck), Synaptophysin (Sigma), Pdgfra (Cell Signaling), Ki67 (Abcam). Tumor grading was determined on the basis of the WHO grading system for malignant astrocytoma.

## Microarray analysis

Three independent NSC lines from each indicated genotype were pooled and submitted for genome wide gene expression profiling using the Illumina Mouse ref 8 v3.0 expression bead chip. Raw data were processed using the 'lumi' package (*Du et al., 2008*) within Bioconductor (*Gentleman et al., 2004*) first by applying the variance-stabilizing transform, and then carrying out quantile normalization. Following this, genes were selected on the basis of a 1.5-fold threshold on the mean of the technical duplicates (technical correlation being higher than 0.996 in all cases). The microarray data depicted in *Figure 3A* has been uploaded to the GEO database under the accession number GSE76296.

Using our normalised microarray data, we calculated the average log fold change between each condition and the wild-type samples. We then calculated distance (following the procedure of [*Dabney, 2006*]) of these expression profiles to the centroids of the four GBM subtypes identified in (*Verhaak et al., 2010*). We mapped the expression profile from mouse to human by matching up gene names between the two species, resulting in a correspondence for 674 of the 840 genes in the GBM signature. To assess informally the specificity of the subtypes, and to gain some insight into the strength of the similarities to GBM subtypes, we bootstrapped the log fold changes within samples 1000 times, recording the pairwise distances between bootstrapped samples and fixed GBM subtype-centroids.

## Copy number variation calling

Total genomic DNA was extracted and purified using DNeasy Blood & Tissue Kit (Qiagen). Copy number variation (CNV) detection was performed using the Affymetrix Mouse Diversity Genotyping Array (MDGA) at AROS Applied Biotechnology. Genotypes were called from the CEL files using the BRLMM-P algorithm in Affymetrix Power Tools (APT) (available from: http://www.affymetrix.com/estore/partners_programs/programs/developer/tools/powertools.affx) using default settings, including quantile normalization. PennAffy (available from: http://penncnv.openbioinformatics.org/en/latest/user-guide/download/) was implemented to generate Log R Ratio (LRR) and B-allele frequency (BAF) using canonical genotype clustering and population frequency of the B-allele (PFB) files kindly provided by Locke et al. (*Locke et al., 2015*). Markers that do not probe allelic balance (exon markers in the MDGA) were removed. GC content was calculated for 1 Mb windows centred by each marker using BEDTools (*Quinlan and Hall, 2010*). LRR was then corrected for genomic waviness by subtracting the median normalised GC content multiplied by a coefficient optimised to generate the minimum variance when subtracted from the LRR. Copy number states were then called for mouse autosomes using the 'CGHcall' R package (*van de Wiel et al., 2007*) in Bioconductor (*Gentleman et al., 2004*) using the 'sdundo' option for the undo.splits parameter for segmentation. Only CNVs larger than 1 Mb were reported.

## TCGA data analysis

Gene expression measurements were obtained by downloading the 'Glioblastoma Multiforme (TCGA, provisional)' dataset using the cBioPortal (http://www.cbioportal.org/, version 05/09/2014) for The Cancer Genome Atlas (http://cancergenome.nih.gov/). The dataset contained 500 microarrays and 153 RNA-seq measurements for 518 glioblastoma patients. Glioblastoma expression subgroup information was obtained from (*Brennan et al., 2013*), and gene expression levels were represented as z-scores. The RNA-seq and microarray data were analyzed separately but the z-scores for *PDGFRA* are correlated across 135 samples for which both data types are available (Spearman correlation coefficient = 0.85 for 135 samples with both RNA-seq and microarray data, *Figure 6—figure supplement 2*), indicating that the two measurement types are comparable. Differences in expression z-scores between *TP53* mutant and *TP53* wt status, as well as glioblastoma subgroups (combining GCiMP and Proneural versus Neural, Classical, and Mesenchymal) were tested using the Wilcoxon's test. Three outlier points from each of the TP53 wt and mutant groups in *Figure 6D*, as well as six outlier points (5 Proneural and 1 Mesenchymal) in *Figure 6E* are excluded from the plot for clarity.

## Statistical analysis

Tumor-free survival was analyzed using GraphPad Prism 6 and statistical analyses to determine tumor-free survival were performed using the Mantel-Cox test. For TCGA data analysis p values were determined using the Wilcoxon's test. For all other experiments with error bars, the unpaired Student's t-test was performed to determine statistical significance.

## Acknowledgements

We thank E Sahai for critical reading of the manuscript and C Cremona and E Schröder for assistance in editing the manuscript. We are indebted to E Nye, B Spencer-Dene, and the Experimental Histopathology Lab at the Crick at Lincoln's Inn Field for technical assistance and Mike Howell from the High Throughput Screening Laboratory for IncuCyte assistance. We thank Elizabeth Locke at the Department of Computer Science, University of Western Ontario, for providing us with files required for processing our Mouse Diversity Genotyping array data and for analysis support and advice.

## Additional information

### Funding

| Funder | Grant reference number | Author |
|---|---|---|
| Cancer Research UK | Core funding | Axel Behrens |
| European Research Council | 281661 ATMINDDR | Axel Behrens |
| Federation of European Biochemical Societies | Long-term fellowship | Sophia M Blake |
| European Molecular Biology Organization | Long-term fellowship | Sophia M Blake Stefan H Stricker |
| Cancer Research UK | Postdoctoral fellowship | Sophia M Blake Nnennaya Kanu |
| Breast Cancer Research Foundation | Grant | Nnennaya Kanu |
| Medical Research Council | G0800020.85704 | Silvia Marino |
| Epigenesys | FP7 Network of Excellence | Anna R Poetsch |
| Peter and Traudl Engelhorn Foundation | Fellowship | Anna R Poetsch |
| The Brain Tumor Charity | Grant | Steven M Pollard |
| Alex Bolt | Research Fellowship | Steven M Pollard |

The funders had no role in study design, data collection and interpretation, or the decision to submit the work for publication.

## Author contributions

SMB, Conceived and designed study, Acquired data, Analyzed and interpreted data, Drafted the manuscript; SHS, Generated data from human glioma lines and performed intra-cranial brain injections, Assisted with study design, Conceived and designed study, Acquired data, Analyzed and interpreted data, Drafted the manuscript; HH, Assisted with data acquisition; ARP, NML, Analyzed TCGA data; GC, Analysed SNP array data; GK, Analysed the mouse microarray data and performed the mouse:GBM-subtype analysis; NK, Generated conditional Atmin mouse; SM, Analyzed histopathology of the tumors; SMP, Performed intra-cranial brain injections, Assisted with the design; AB, Supervised experiments and contributed as senior author, Drafting of the manuscript

## Ethics

Animal experimentation: Experiments in mice were carried out in accordance with the UK's Animals (Scientific Procedures) Act 1986, with the approval of the Crick Institute's Ethical Review Committee and under the guidance of the Biological Resources Unit.

# Additional files

## Supplementary files

• Supplementary file 1. Causes of death for experimental cohorts in *Figures 1* and *2*.

• Supplementary file 2. Primer sequences.

## Major datasets

The following datasets were generated:

| Author(s) | Year | Dataset title | Dataset URL | Database, license, and accessibility information |
|---|---|---|---|---|
| Behrens A, Blake S | 2015 | ATMIN function in p53-deficient GBM | http://www.ncbi.nlm.nih.gov/geo/query/acc.cgi?acc=GSE76296 | Publicly available at the NCBI Gene Expression Omnibus (Accession no: GSE76296) |

The following previously published dataset was used:

| Author(s) | Year | Dataset title | Dataset URL | Database, license, and accessibility information |
|---|---|---|---|---|
| Glioblastoma Multiforme | 2014 | Glioblastoma Multiforme (TCGA, provisional) | http://cancergenome.nih.gov/ | Publically available |

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
