## [Decision Letter]

Thank you for submitting your work entitled "Inactivation of the ATMIN/ATM pathway protects against glioblastoma formation" for peer review at *eLife*. Your submission has been favorably evaluated by Sean Morrison (Senior editor), a Reviewing editor, and three reviewers.

The reviewers have discussed the reviews with one another and the Reviewing editor has drafted this decision to help you prepare a revised submission.

This manuscript describes an interesting pro-tumorigenic role for ATMIN mediated ATM signaling in glioblastoma. Specifically, the authors demonstrate that the concomitant loss of ATMIN and p53 in nestin positive cells in mice, reduces glioma formation. This result is recapitulated in a transplant model using neural stem cells. The authors conclude that p53 drives glioma, and that the resulting tumors activate PDGFRa through an ATM and ATMIN dependent pathway.

In order for us to consider your manuscript further, please pay close attention to the following critiques.

Essential revisions:

1) Comment on the potential importance of ROS in the transforming mechanism described in the paper.

2) The authors should provide the complete microarray data in an appropriate format.

3) Demonstrate that ATMi and *Pdgfra* were was within a reasonable dose range, clarify dose and dose response against ATM and PDGFRa phosphorylated substrates, to assure that effects seen were on rather than off-target.

4) Explain why, if ATMIN sub serves ATM downstream signaling as its primary output, deletion of ATMIN (Figure 2) was more effective than deletion of ATM (Figure 4) in reducing tumorgenicity.

5) Discuss how ATM and ATMIN contribute to elevated *Pdgfra* signaling in this model.

6) The conclusion that alterations observed in experimental gliomas are real tumors with secondary genetic changes requires characterizing the genetic status of the tumors by SNP or CGH arrays.

*Reviewer #1:*

This manuscript describes an unexpected pro-tumorigenic role for ATMIN mediated ATM signaling in glioblastoma using a variety of mouse and human models. The deletion of both ATM and ATMIN in mice, and in the case of ATM, humans, leads to the development of different types of tumors, primarily of lymphocyte origin. Thus both ATM and ATMIN are primarily regarded as tumor suppressors. Recent work has established that ATM may have some pro-tumorigenic roles through its functions in differentiation and secretory phenotypes in different tumor models (Santos et al., Nature 2014 and Chen et al., *ELife* 2015).

The authors demonstrate that the concomitant loss of ATMIN in mice where p53 has been deleted by Nestin-Cre reduces tumor formation that is primarily of glioblastoma origin. This result is convincingly recapitulated in a transplant model using NSCs. The authors identified PDGFRa as a gene strongly affected by ATMIN or ATM status in the p53∆ tumors and demonstrated that its levels affected cell growth through a variety of complementary experiments. Finally, the authors demonstrate that combined inhibition of both ATM and PDGFR increases cell death and decreases proliferation in human glioblastoma cell lines with elevated PDGFRa and mutant p53.

The work is generally of high quality and will be of interest to numerous fields for the following reasons: 1) This adds to a growing body of evidence that ATM signaling can have both pro and anti-tumorigenic roles, and thus provide information regarding the appropriate use of ATM inhibitors; 2) It establishes in some further detail that p53-Nestin-Cre deletion is a useful glioblastoma model, a point that I think is not widely appreciated; 3) It identifies PDGFRa as a downstream player in ATM/ATMIN signaling that may also be relevant in other contexts; and 4) It establishes some proof of principal that ATM inhibitors may be useful to combine with PDGFRa depletion in p53 mutant glioblastomas. While there are some mechanistic details that are not well established, primarily the mechanism by which ATM upregulates PDGFRa and the trigger for ATMIN/ATM activation in the p53 null tumors, the establishment of all of these details would likely not have a strong impact on the primary conclusions of the paper.

Comments:

1) In the last paragraph of the subsection “*Pdgfra* overexpression is reversible with pharmacological ATM inhibition”, the authors conclude that "*Pdgfra* upregulation is not a consequence of stimulation of DNA damage signaling" based on the observation that HU or IR do not upregulate *Pdgfra* levels, and the lack of differences in some ATM substrates. Previous work from this group and others (Kanu and Behrens, EMBO J. 2007 and Jurado et al. 2010) has made the point that ATMIN does not seem to play a prominent role in the activation of ATM in response to IR or HU but is a major player in the response to reactive oxygen. The inhibitor experiments argue that ATM activity is required and it is thus presumably activated in the presence of ATMIN. Given that ROS is linked to the activation of ATM by ATMIN, p53 signaling (Liu et al., Free Radical Biol Med (review) 2008) and implicated in PDGFRa activity (Lei and Kazlauskas, MCB 2013), it seems that this would be an important stimuli to test before making this very definitively stated conclusion.

2) The authors should provide the complete microarray data, as there are a number of other genes altered by ATMIN depletion that may be of interest for understanding the phenotypes (possibly factors induced by ROS for example?) and potential impact on therapy.

*Reviewer #1 (Additional data files and statistical comments):*

As indicated in the previous comments, the microarray data should be supplied (and deposited to the appropriate public database) and additional data regarding the numbers of technical or biological replicates used in any of the statistical analyses should be clearly stated.

*Reviewer #2:*

This is an interesting if somewhat preliminary manuscript that argues, based initially on in vivo mouse modeling, that loss of p53 drives glioma, and that the resulting tumors activate PDGFRa through an ATM and ATMIN dependent pathway. While this pathway is new, no mechanism is revealed for how p53 signals through ATM to drive PDGFRA. In addition, the authors argue based solely on cell line data, for enhanced effects when combining PDGFR and ATM inhibitors for preclinical therapies.

1) Can the authors subclassify mouse and human p53 mutant GBM based on TCGA classification? *Pdgfra* is enriched in proneural GBM. It would be important to know whether p53 mt murine GBM are proneural. This is also relevant for Figure 6. What is TCGA classification of GNSCs used, are these also proneural? If not, do the authors think that this mechanism is relevant for all subtypes?

The authors should show that ATMi and *Pdgfra* used was within a reasonable dose range, clarify dose and dose response against ATM and PDGFRa phosphorylated substrates, to assure that effects seen were on rather than off-target.

If ATMIN sub serves ATM downstream signaling as its primary output, why was deletion of ATMIN (Figure 2) more effective than deletion of ATM (Figure 4) in reducing tumorgenicity.

The authors show that induction of *Pdgfra* in p53mt NSCs is not a consequence of DNA damage signaling. So how do ATM and ATMIN contribute to elevated *Pdgfra* signaling in this model?

The result of PDGFRA depletion is only shown in a single human cell line.

As the authors mention, PDGFRi has, so far, failed transition to clinical practice. Is the effect, and particularly the apoptosis observed in vitro here, translatable to an in vivo setting? The authors argue that combined inhibition of ATM and PDGFR promises new opportunities for future GBM treatment. The addition of in vivo experiments is critical here to substantiate this statement, and to determine whether this effect really drives apoptosis, and with it, regression of established tumors.

Reviewer #2 (Additional data files and statistical comments):

No issues for me, except to wonder about differences between loss of ATMIN and loss of ATM, listed above.

*Reviewer #3:*

The paper "Inactivation of the ATMIN/ATM pathway protects against glioblastoma formation" reports studies in a *Nestin-Cre (p53^ΔΝ^)* model in which the activation of the gene ATMIN is identified as an important event downstream to p53 deletion to implement the tumorigenic phenotype in the brain. Mechanistically, the authors suggest that ATMIN is required to sustain the high expression of PDGFRA in this model.

The development of a new, albeit predictable model of brain tumors obtained by deleting p53 in the neural stem cell compartment is a potentially interesting finding. However, the role played by ATMIN in the tumorigenic process and especially the suggestion that ATMIN is a factor required for the overexpression of PDGFRA remain of uncertain significance. Conversely, it seems that the findings reported are limited to the mouse model under investigation and unlikely to have an impact for the biology of human glioma. Indeed, the main weakness of the manuscript is that in human glioma the high expression of PDGFRA (in the presence or absence of p53 mutations) occurs through gene amplification and it is therefore unlikely that ATMIN plays any role for PDGFRA overexpression. Furthermore, there are no known genetic alterations reported to target ATMIN in human malignant glioma.

Therefore, while the phenotypic alterations characterized by the authors have relevance in the mouse model under investigation, there is no indication that the information contained in this manuscript has any relevance for the human disease.

Specific points:

1) Although the authors perform a phenotypic characterization of the brain lesions occurring in the *Nestin-Cre (p53^ΔΝ^)* mouse, the conclusion that the alterations they observe are real tumors with secondary genetic changes requires characterizing the genetic status of the tumors by SNP or CGH arrays.

2) PDGFRA is amplified and overexpressed in the subgroup of human GBM defined as proneural. Do the *Nestin-Cre (p53^ΔΝ^)* tumors belong to this molecular subgroup?

3) The authors state that p53 alterations are early events, implicated in GBM initiation. It is not clear where the authors obtain this information, especially in the context of the human disease.

4) In Figure 2, hypoxia has only a minor ability to induce apoptosis if compared with the effects produced by IR. Therefore, the rescue of this effect by loss of ATMIN is similarly of limited significance.

5) The authors show that the integrity of the ATMIN gene is essential to preserve overexpression of PDGFRA in the *Nestin-Cre (p53^ΔΝ^)* mouse model. However, this leaves open the question of whether PDGFRA overexpression is sufficient to reverse the effects of ATMIN deletion. The authors should address this question through reconstitution of the expression of PDGFRA in *Nestin-Cre (p53^ΔΝ^) Atmin^ΔΝ^* mouse.

---

## [Author Response]

Essential revisions:

*1) Comment on the potential importance of ROS in the transforming mechanism described in the paper.*

ROS is an important regulator of ATM signalling. ATM can be activated by ROS by the introduction of a disulfide bond (Guo et al. Science 330, 517-521, 2010), and under certain oxidising conditions ATM function requires ATMIN (Kanu et al. J Biol Chem 285, 38534–38542, 2010). Moreover, ROS levels are increased in ATM mutant cells, and strikingly, thymomas in ATM mutant mice can be significantly delayed by anti-oxidants treatment (Reliene et al. 2007).

Importantly, constitutive “baseline” ATM/ATMIN signaling is required for *pdgfr* regulation, and what we view as “baseline” could actually represent ATM signalling induced by endogenous ROS. We have now amended the Results section to specify that the DNA damage signalling we refer to at that point is that induced by IR and HU, not ROS.

Results section:

“This suggests that while ATM is required for the increased *Pdgfra* expression in *p53^ΔN^* cells, this does not involve alterations in IR or HU-induced ATM signaling, but at this point does not exclude the possibility that increased ROS (reactive oxygen species) levels in p53^∆N^ GBM cells might contribute to ATM pathway activation and subsequent *Pdgfra* induction and GBM development.”

*2) The authors should provide the complete microarray data in an appropriate format.*

The microarray data depicted in Figure 3 has been uploaded to the GEO database under the accession number GSE76296 and made publically available (the data can be viewed at http://www.ncbi.nlm.nih.gov/geo/query/acc.cgi?acc=GSE76296). This information is now also mentioned in the corresponding Materials and methods section.

**“**Microarray analysis:

Three independent NSC lines from each indicated genotype were pooled and submitted for genome wide gene expression profiling using the Illumina Mouse ref 8v3 expression bead chip. […] The microarray data depicted in Figure 3 has been uploaded to the GEO database under the accession number GSE76296.”

*3) Demonstrate that ATMi and Pdgfra were was within a reasonable dose range, clarify dose and dose response against ATM and PDGFRa phosphorylated substrates, to assure that effects seen were on rather than off-target.*

We have performed dose titration experiments in GNSCs for both the ATM and PDGFRa inhibitors and can confirm that the concentrations chosen for the experiments were the minimally required concentrations to achieve target and downstream pathway inhibition(Figure 7—figure supplement 1).

ATM: We used 10µM of ATMi in our experiments. GNSCs were treated with DMSO or 400nM, 2µM or 10µM of ATMi for 16 hrs, then subjected to 5Gy of ionizing irradiation and harvested 30 minutes later. Pre-treatment of cells with the ATMi at 10µM clearly prevented auto-phosphorylation of ATM at Serine residue 1981, indicative of ATM activation, and the phosphorylation of the well-characterized ATM substrates Kap1 and Chk2, whereas the lower tested doses were not able to fully inhibit substrate phosphorylation. This shows that the applied concentration of the inhibitor in the performed experiments efficiently inhibited ATM activity.

For PDGFRa inhibition, we used 1µM of inhibitor in our experiments. To test PDGFRa activity, GNSCs were treated with DMSO, 200nM or 1µM of PDGFRa inhibitor for 2hrs. PDGFRa activation was measured by IP of PDGFRa, followed by western blot for phosphorylated *Pdgfra* tyrosine residue 754, a hallmark of *Pdgfra* activation. 1µM inhibitor clearly reduced phosphorylation of *Pdgfra* at Tyrosine residue 754 and also reduced the phosphorylation of the downstream substrate Akt at Serine residue 473. Hence, 1µM, but not 200nM PDGFRa inhibitor, blocks substrate and downstream target phosphorylation/signaling.

*4) Explain why, if ATMIN sub serves ATM downstream signaling as its primary output, deletion of ATMIN (Figure 2) was more effective than deletion of ATM (Figure 4) in reducing tumorgenicity.*

Although the primary output of ATMIN signaling is mediated by the upstream kinase ATM, it is also clear that ATMIN has additional, ATM independent, functions. ATMIN has been shown to bind chromatin and act as transcriptional regulator in several cell types (Goggolidou et al., 2014; Jurado et al., 2012; Jurado et al., 2011). Therefore, in addition to sub-serving ATM, ATMIN also has ATM independent functions. A single one, or more likely, a combination of these ATMIN target genes may be responsible for the enhanced anti-tumorigenic effect observed.

We now discuss this in the Results section:

“However, ATMIN deletion (Figure 2) was more effective than deletion of ATM (Figure 4). ATMIN can also act as a transcription factor (Jurado et al., 2012; Goggolidou et al., 2014; Jurado et al., 2012), and it is conceivable that this ATM-independent function of ATMIN contributes to the suppression on GBM.”

*5) Discuss how ATM and ATMIN contribute to elevated Pdgfra signaling in this model.*

Analysis of published ENCODE data sets failed to reveal binding of p53 to the *PDGFR* promoter. Similarly, our in house ChIP-Seq experiments showed no ATMIN occupancy on the *PDGFR* promoter; Kanu et al., unpublished). Therefore the antagonistic regulation of *PRGFR* expression by p53 and ATM/ATMIN signalling must be indirect.

We speculate that p53 controls the expression of a negative regulator of *PDGFR*, e.g. a repressor or a miR RNA, and that ATM signalling is required for the expression or function of this factor. We have discussed this in the revised manuscript.

As the microarray data reveals 39 up-regulated and 41 repressed genes specifically altered in the double mutant cells vs. the p53 single mutants, it will require a designated pooled shRNA/CRISPR screen to confirm or de-validate this hypothesis. We aim to perform these experiments in follow up projects.

Discussion section:

“In our murine and human glioma lines *Pdgfra* levels could be dynamically modulated by depleting ATMIN or inhibiting ATM. […] As these intermediate factors could be promising targets for pharmacological inhibition, further investigation into their identity is of therapeutic interest.”

*6) The conclusion that alterations observed in experimental gliomas are real tumors with secondary genetic changes requires characterizing the genetic status of the tumors by SNP or CGH arrays.*

This is an excellent comment from the reviewer and we have addressed this question by conducting SNP array analysis for primary cells derived from 5 *p53^ΔN^* gliomas. One adult *wt* NSCs as well as 3 non-tumorigenic adult *p53^ΔN^* NSCs were used as controls. The results from the SNP array are depicted in (Figure 1—figure supplement 3 and Figure 1—figure supplement 4) and referenced in the Results section of the manuscript. The SNP array analysis was performed on Affymetrix® Mouse Diversity Genotyping arrays and showed, that no whole chromosome copy number variations or copy number variations larger than the defined detection limit (> 1 Mb) were found in *wt* NSCs or p5*3^ΔN^*NSCs, confirming that these cells have not developed secondary alterations. In contrast, large chromosomal gains and losses (typical tumorigenic genetic changes) were found in 4 out of 5 analysed *p53^ΔN^*glioma cell lines, confirming the presence of secondary genetic alterations in these cell lines, while not excluding the further presence of point mutations, insertions/deletions that are smaller than 1 Mb.

Results section:

“The long latency of these tumors, together with the genome instability and transcriptional changes known to be induced by loss of p53, indicated that these gliomas almost certainly develop as a result of secondary mutations arising in the p53-null brain. In line with this notion, SNP array revealed large chromosomal gains and losses (typical tumorigenic genetic changes) in 4 out of 5 analysed *p53^ΔN^* gliomas whereas no changes were detected in 1 wt and 3 p53-deficient non-tumorigenic control NS cells.”

Reviewer #1: Comments:

*1) In the last paragraph of the subsection “Pdgfra overexpression is reversible with pharmacological ATM inhibition”, the authors conclude that "Pdgfra upregulation is not a consequence of stimulation of DNA damage signaling" based on the observation that HU or IR do not upregulate Pdgfra levels and the lack of differences in some ATM substrates. Previous work from this group and others (Kanu and Behrens, EMBO J. 2007 and Jurado et al. 2010) has made the point that ATMIN does not seem to play a prominent role in the activation of ATM in response to IR or HU but is a major player in the response to reactive oxygen. The inhibitor experiments argue that ATM activity is required and it is thus presumably activated in the presence of ATMIN. Given that ROS is linked to the activation of ATM by ATMIN, p53 signaling (Liu et al., Free Radical Biol Med (review) 2008) and implicated in PDGFRa activity (Lei and Kazlauskas, MCB 2013), it seems that this would be an important stimuli to test before making this very definitively stated conclusion.*

We thank the referee for this important comment. And have addressed this comment in our reply to the Editorial comment 1.

*2) The authors should provide the complete microarray data as there are a number of other genes altered by ATMIN depletion that may be of interest for understanding the phenotypes (possibly factors induced by ROS for example?) and potential impact on therapy.*

We now provide the complete microarray data, as described in our response to the Editorial comment 2.

*Reviewer #1 (Additional data files and statistical comments): As indicated in the previous comments, the microarray data should be supplied (and deposited to the appropriate public database) and additional data regarding the numbers of technical or biological replicates used in any of the statistical analyses should be clearly stated.*

In response to this reviewer’s comment and the Editorial comment 2, we have deposited the microarray data in the GEO database with the accession number GSE76296. In addition, the data regarding the numbers of technical or biological replicates used for the statistical analyses are now included in the figure legends of the revised manuscript.

*Reviewer #2: This is an interesting if somewhat preliminary manuscript that argues, based initially on in vivo mouse modeling, that loss of p53 drives glioma, and that the resulting tumors activate PDGFRa through an ATM and ATMIN dependent pathway. While this pathway is new, no mechanism is revealed for how p53 signals through ATM to drive PDGFRA. In addition, the authors argue based solely on cell line data, for enhanced effects when combining PDGFR and ATM inhibitors for preclinical therapies. 1) Can the authors subclassify mouse and human p53 mutant GBM based on TCGA classification? Pdgfra is enriched in proneural GBM. It would be important to know whether p53 mt murine GBM are proneural. This is also relevant for Figure 6. What is TCGA classification of GNSCs used, are these also proneural? If not, do the authors think that this mechanism is relevant for all subtypes?*

To address this valid point, we compared our data to the TCGA classification of human GBM as described in (Verhaak et al., 2010).This analysis revealed that the only transcriptional profile that significantly resembles any of the human TCGA defined subtypes, is the one derived from *p53^ΔN^*NSCs. These cells show close relation to the human pro-neural subtype. We now report this observation in the Results section of the manuscript and have added the corresponding data analysis as a new supplementary figure (Figure 3—figure supplement 1).

Results section:

**“**Increased PDGFRa levels are also a characteristic hallmark of the proneural GBM subtype as classified in (Brennan et al., 2013; Verhaak et al., 2010). Interestingly we found that *p53^ΔN^* NSCs were associated most closely with the proneural subtype when comparing the microarray expression profile from the murine NSCs to that of publically available human GBM samples from TCGA (Verhaak et al., 2010) (Figure 3—figure supplement 1), which is in agreement with elevated *Pdgfra* mRNA as well as protein levels (Figure 3).”

*The authors should show that ATMi and Pdgfra used was within a reasonable dose range, clarify dose and dose response against ATM and PDGFRa phosphorylated substrates, to assure that effects seen were on rather than off-target.*

We have addressed this important suggestion and shown that the chosen dose was required and sufficient to engage the target and display downstream pathway modulation. Please see our response to the Editor’s comments 3 and Figure 7—figure supplement 1.

*If ATMIN sub serves ATM downstream signaling as its primary output, why was deletion of ATMIN (Figure 2) more effective than deletion of ATM (Figure 4) in reducing tumorgenicity.*

ATMIN has a dual function, as an ATM cofactor and as a transcription factor. As discussed in more detail in the response to Editor’s comments 4, we speculate that the function of ATMIN as transcription factor contributes to the pro-tumorigenic effect observed.

*The authors show that induction of Pdgfra in p53mt NSCs is not a consequence of DNA damage signaling. So how do ATM and ATMIN contribute to elevated Pdgfra signaling in this model?*

It is interesting to speculate how p53 and ATMIN/ATM signalling control PDGFRa expression, which has been addressed in more detail in the Editor’s comment 5. In brief, the regulation is most likely indirect, and possibly involves a p53-regulated factor that requires ATM activity for function.

The result of PDGFRA depletion is only shown in a single human cell line.

We agree with the reviewer’s comment about the limited sample number in the PDGFRa knockdown experiments. We generated this Dox-inducible p53-deficent cell line allowing PDGFR depletion to provide a proof of principle validation for PDGFRa function. We have followed this in using 2 different PDGFRa inhibitors, in 3 independent (one wild-type and 2 p53-deficient) human derived neural stem cell lines. As shown in Figure 7 and Figure 7—figure supplement 2 and Figure 7—figure supplement 3 both inhibitors display similar effects. We hope that this is satisfactory to demonstrate the involvement of PDGFRa function in glioma stem cell function.

As the authors mention, PDGFRi has, so far, failed transition to clinical practice. Is the effect, and particularly the apoptosis observed in vitro here, translatable to an in vivo setting? The authors argue that combined inhibition of ATM and PDGFR promises new opportunities for future GBM treatment. The addition of in vivo experiments is critical here to substantiate this statement, and to determine whether this effect really drives apoptosis, and with it, regression of established tumors.

As the reviewer suggests, the combination therapy of ATM and PDGFR inhibitors in a pre-clinical setting is a logical next step. However, this experiment is extremely difficult to perform for two reasons. Firstly, the long latency of gliomas to arise in the p53 F/F; Nestin-Cre strain (around 300 days, Figure 1), does not allow us to perform this experiment within the provided time of a normal revision process. Secondly, the ATM inhibitor has been shown to not cross the blood brain barrier (Biddlestone-Thorpe et al., 2013), essentially precluding the usage of the current generation of ATM inhibitors for this experiment, making it currently impossible to address this valid suggestion.

*Reviewer #2 (Additional data files and statistical comments): No issues for me, except to wonder about differences between loss of ATMIN and loss of ATM, listed above.*

As described in more detail in our response to the Editor’s comment 4, we speculate that this is due to a pro-tumorigenic role of ATMIN-mediated transcription.

*Reviewer #3: The paper "Inactivation of the ATMIN/ATM pathway protects against glioblastoma formation" reports studies in a Nestin-Cre (p53^ΔΝ^) model in which the activation of the gene ATMIN is identified as an important event downstream to p53 deletion to implement the tumorigenic phenotype in the brain. Mechanistically, the authors suggest that ATMIN is required to sustain the high expression of PDGFRA in this model.*

*The development of a new, albeit predictable model of brain tumors obtained by deleting p53 in the neural stem cell compartment is a potentially interesting finding. However, the role played by ATMIN in the tumorigenic process and especially the suggestion that ATMIN is a factor required for the overexpression of PDGFRA remain of uncertain significance. Conversely, it seems that the findings reported are limited to the mouse model under investigation and unlikely to have an impact for the biology of human glioma. Indeed, the main weakness of the manuscript is that in human glioma the high expression of PDGFRA (in the presence or absence of p53 mutations) occurs through gene amplification and it is therefore unlikely that ATMIN plays any role for PDGFRA overexpression. Furthermore, there are no known genetic alterations reported to target ATMIN in human malignant glioma.*

*Therefore, while the phenotypic alterations characterized by the authors have relevance in the mouse model under investigation, there is no indication that the information contained in this manuscript has any relevance for the human disease. Specific points:*

*1) Although the authors perform a phenotypic characterization of the brain lesions occurring in the Nestin-Cre (p53^ΔΝ^) mouse, the conclusion that the alterations they observe are real tumors with secondary genetic changes requires characterizing the genetic status of the tumors by SNP or CGH arrays.*

This is a very valid point and has been addressed in detail in our response to the Editor’s comment 6. The corresponding data is presented in Figure 3 and reported in the Results section of the manuscript.

Results section:

“The long latency of these tumors, together with the genome instability and transcriptional changes known to be induced by loss of p53, indicated that these gliomas almost certainly develop as a result of secondary mutations arising in the p53-null brain. In line with this notion, SNP array revealed large chromosomal gains and losses (typical tumorigenic genetic changes) in 4 out of 5 analysed *p53^ΔN^*gliomas whereas no changes were detected in 1 wt and 3 p53-deficient non-tumorigenic control NS cells.”

*2) PDGFRA is amplified and overexpressed in the subgroup of human GBM defined as proneural. Do the Nestin-Cre (p53^ΔΝ^) tumors belong to this molecular subgroup?*

As mentioned in our response to the first comment of referee 2, we compared our data to the TCGA classification of human GBM as described in (Verhaak et al., 2010).This analysis revealed that the only transcriptional profile that significantly resembles any of the human TCGA defined subtypes, is the one derived from *p53^ΔN^*NSCs, and these cells show close relation to the human pro-neural subtype. We now report this observation in the Results section of the manuscript and have added the corresponding data analysis as a new supplementary figure (Figure 3—figure supplement 1).

Results section:

**“**Increased PDGFRa levels are also a characteristic hallmark of the proneural GBM subtype as classified in (Brennan et al., 2013; Verhaak et al., 2010). Interestingly we found that *p53^ΔN^* NSCs were associated most closesly with the proneural subtype when comparing the microarray expression profile from the murine NSCs to that of publically available human GBM samples from TCGA (Verhaak et al., 2010) (Figure 3—figure supplement 1), which is in agreement with elevated *Pdgfra* mRNA as well as protein levels (Figure 3).”

*3) The authors state that p53 alterations are early events, implicated in GBM initiation. It is not clear where the authors obtain this information, especially in the context of the human disease.*

We are sorry for not pointing this out more clearly. The articles cited in the main manuscript (Maher et al., 2001; Wang et al., 2009; Ohgaki et al., 2004), provide evidence to studies conducted in human patients as well as mouse models, showing that loss of p53 function is observable in both primary and secondary GBMs, concluding p53 loss to be an early event during gliomagenesis. We have now added an additional 2 references to the manuscript to further substantiate this claim.

*4) In Figure 2, hypoxia has only a minor ability to induce apoptosis if compared with the effects produced by IR. Therefore, the rescue of this effect by loss of ATMIN is similarly of limited significance.*

Compared to the IR induced cell death the effect of hypoxia on cell survival is indeed less pronounced. We would however like to stress that hypoxia is a hallmark of GBM tumours (e.g. Evans et al., 2004) what in our view warrants the inclusion of these data in the manuscript.

*5) The authors show that the integrity of the ATMIN gene is essential to preserve overexpression of PDGFRA in the Nestin-Cre (p53^ΔΝ^) mouse model. However, this leaves open the question of whether PDGFRA overexpression is sufficient to reverse the effects of ATMIN deletion. The authors should address this question through reconstitution of the expression of PDGFRA in Nestin-Cre (p53^ΔΝ^) Atmin^ΔΝ^ mouse.*

We agree with the reviewer that this indeed would be a very interesting point to address. However, as the tumor latency period of *p53^ΔN^* animals is approximately 1 year, and the cross of transgenic PDGFRa mice to the triple transgenic genotype would also require approx. 1 year, this experiment is unfortunately not within the scope of the study.